# PROVENCE: EFFICIENT AND ROBUST CONTEXT PRUNING FOR RETRIEVAL-AUGMENTED GENERATION

**Nadezhda Chirkova, Thibault Formal, Vassilina Nikoulina, Stéphane Clinchant**
Naver Labs Europe
Grenoble, France
`{nadia.chirkova,thibault.formal}@naverlabs.com`

## ABSTRACT

Retrieval-augmented generation improves various aspects of large language models (LLMs) generation, but suffers from computational overhead caused by long contexts as well as the propagation of irrelevant retrieved information into generated responses. Context pruning deals with both aspects, by removing irrelevant parts of retrieved contexts before LLM generation. Existing context pruning approaches are however limited, and do not provide a universal model that would be both *efficient* and *robust* in a wide range of scenarios, e.g., when contexts contain a variable amount of relevant information or vary in length, or when evaluated on various domains. In this work, we close this gap and introduce `Provence` (Pruning and Reranking Of retrieVEd relevaNt ContExts), an efficient and robust context pruner for Question Answering, which dynamically detects the needed amount of pruning for a given context and can be used out-of-the-box for various domains. The three key ingredients of `Provence` are formulating the context pruning task as sequence labeling, unifying context pruning capabilities with context reranking, and training on diverse data. Our experimental results show that `Provence` enables context pruning with negligible to no drop in performance, in various domains and settings, at almost no cost in a standard RAG pipeline. We also conduct a deeper analysis alongside various ablations to provide insights into training context pruners for future work. Our model[1] and code[2] are publicly available.

## 1 INTRODUCTION

Retrieval-Augmented Generation (RAG) has become a widely-used paradigm for improving factuality, attribution, and adaptability of Large Language Models (LLMs) (Das et al., 2019; Asai et al., 2024; Seo et al., 2019; Lewis et al., 2020; Mallen et al., 2023a; Min et al., 2023). Augmenting a given user's query with retrieved relevant contexts helps to avoid the generation of untruthful information and enables the provision of references used to generate the answer. Furthermore, using a domain-specific datastore may enable access and reasoning over a previously unknown knowledge – without fine-tuning the LLM. One additional advantage of the RAG approach is the easy plug-and-play architecture (LangChain): practitioners may choose components (retrievers, generator LLMs, context granularity etc.) which best suit their particular cases to maximize the final performance.

At the same time, the use of RAG adds *computational overhead* due to both retrieval latency and the increased input length for the LLMs. It may also propagate *irrelevant information* present in retrieved contexts into generated responses. These issues can be solved by developing more efficient and robust LLMs – either by making architectural changes to process long contexts more efficiently (Nawrot et al., 2024; Dao, 2024; Chevalier et al., 2023) or increasing the diversity of the tuning data to improve processing of irrelevant contexts (Lin et al., 2024). However, tuning the LLM can be highly resource-consuming, or even impossible to apply for proprietary (closed) LLMs. An alternative solution consists in *pruning retrieved contexts* by removing context parts irrelevant to the user's query – which reduces context lengths and therefore speeds up generation. Such context

---

[1]Our model: `https://huggingface.co/naver/provence-reranker-debertav3-v1`
[2]Our code: `https://github.com/naver/bergen/tree/main/scripts/provence`

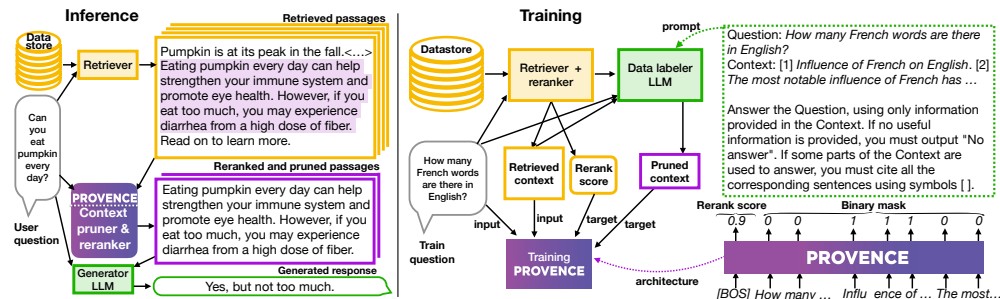

Figure 1: Illustration of inference (*left*) and training (*right*) of `Provence`.

Table 1: Analysis of existing approaches for context pruning. Violet / Orange highlight practical / less-practical solutions.

| Approach | Query-dep. | Granularity | Output | Type | Base arch. | Multi-domain testing | Model release |
|---|---|---|---|---|---|---|---|
| Selective Context | No | token-level | % of tokens | extr. | Llama-7B / GPT2 | Yes | Yes |
| LLMLingua | No | token-level | % of tokens | extr. | Alpaca-7B / GPT2 | Yes | Yes |
| LongLLMLingua | Yes | token-level | % of tokens | extr. | Llama-2-7B-chat | Yes | Yes |
| LLMLingua2 | No | token-level | % of tokens | extr. | RoBERTa / mBERT | Yes | Yes |
| RECOMP extr. | Yes | sent.-level | $k$ sentences | extr. | BERT | No | Yes |
| RECOMP abstr. | Yes | sent.-level | $\geqslant 0$ sentences | abstr. | T5-L | No | Yes |
| FilCo | Yes | sent.-level | 1 sentence | abstr. | T5-XL / Llama-2-7B | No | No |
| COMPACT | Yes | sent.-level | $\geqslant 0$ sentences | abstr. | Mistral-7B | No | Yes |
| `Provence` (ours) | Yes | sent.-level | $\geqslant 0$ sentences | extr. | DeBERTa | Yes | Yes |

pruning module can be used in a *plug-and-play manner with any generator LLM*, featuring both easy use and better transparency in the RAG pipeline.

Despite initial efforts on developing context pruners for RAG, none of the existing solutions provide a model ready to be used ***out-of-the-box*** in practice. First, many approaches are designed for a simplified setting, e.g., with the assumption that only one sentence per context is relevant to the input query (Wang et al., 2023; Xu et al., 2024), or that the compression ratio is fixed (Jiang et al., 2023; Pan et al., 2024). However, in practice contexts may contain *various portions of relevant information*, from empty to full relevant context, and pruners should detect it in an ***adaptable*** fashion. Second, many works introduce context pruners that are not efficient enough to be used in practice. This includes using billion-sized LLMs as base models for pruners (Jiang et al., 2024; Pan et al., 2024; Wang et al., 2023), or designing abstractive context compressors which require sequential autoregressive generation of the final context (Wang et al., 2023; Xu et al., 2024). We argue that a more practical and ***efficient*** setting consists in fine-tuning a *small-size model* such as DeBERTa (He et al., 2021b;a), as an *extractive* pruner, i.e., with a lightweight prediction head for selecting relevant context parts. Third, most of the existing works train context pruners for each dataset individually and do not target nor test pruners ***robustness*** to various data domains.

Table 1 summarizes the properties of various existing methods along specified dimensions and shows that none of them satisfy all listed criteria. The table also includes a dimension of pruning granularity, i.e., token-level *vs* sentence-level pruning. In this work, we focus on *query-dependent sentence-level* pruning, which prunes out semantic units (sentences) that are deemed not relevant to generate the answer. An alternative approach is token-level pruning which prunes out low-level grammatical units such as articles or interjections, usually in a query-independent fashion. The two approaches are orthogonal and could potentially be combined.

To address listed limitations, we introduce `Provence` (Pruning and Reranking Of retrieVEd relevaNt ContExt), an approach for training an ***adaptable***, ***efficient*** and ***robust*** sentence-level context pruner for Question Answering, which can be used ***out-of-the-box*** across various domains and settings. To achieve this, we formulate context pruning as *binary sequence labeling* so that the binary mask predicted by the pruner determines *sentences* (from zero to all) which are relevant to the query,

and train our pruner from a lightweight DeBERTa model on diverse data. Furthermore, we notice that context pruning and reranking (i.e., the second step in effective retrieval pipelines) bear a strong resemblance. We therefore propose to **unify these two models into a single one**, completely **eliminating the cost** of context pruning in the RAG pipeline.

More specifically, our contributions are as follows:

- We propose an approach for training an *adaptable*, *robust*, and *efficient* context pruner for QA – and release our trained model and code. Three key ingredients of our approach are formulating context pruning as sequence labeling, unifying context pruning and reranking in a single model, and training on diverse data.

- We test `Provence` on various QA domains and show its *out-of-the-box applicability* to prune contexts with negligible to no drop in performance and at almost no cost, substantially outperforming baseline approaches. We also demonstrate `Provence` capabilities in detecting the number of relevant sentences at any positions in the context and robustness to various context lengths.

- We conduct multiple ablations to demonstrate which techniques are essential for training robust context pruners, to provide insights for future context pruners development.

**Definitions.** A typical RAG pipeline consists of *(0)* a user's question, or query; *(1)* a *datastore*, i.e., a collection of *documents* (pieces of text) to be retrieved from, *(2)* an efficient retriever which enables fast retrieval from a large datastore (typically a dual-encoder model, where queries and passages are encoded independently), *(3)* a more expensive cross-encoder reranker which further reduces and reorders a set of retrieved passages (cross-encoding means encoding a passage together with a query); and *(4)* a generator LLM which outputs the final response based on the user's query and the relevant passages. Such a pipeline can be represented as `retrieve >> rerank >> generate`. Context pruning can be incorporated before generation, i.e., `retrieve >> rerank >> prune >> generate`. In our work, we also propose to incorporate context pruning into reranking, an essential and already present component in RAG (Rau et al., 2024a): `retrieve >> rerank+prune >> generate`. This enables **context pruning at almost zero cost**.

## 2 RELATED WORK

**Context pruning.** RECOMP (Xu et al., 2024) focuses on context pruning for RAG and proposes both extractive and abstractive context pruners. The extractive RECOMP approach independently encodes sentences in the context and then selects top sentences with embeddings closest to the query embedding. Such an approach limits context understanding, due to independent processing of both sentences and queries. The method also requires specifying the amount of sentences to keep as a hyperparameter – which is usually unknown in practice and should depend on each particular passage. The abstractive RECOMP summarizes key information from the passage relevant to the query (including zero relevant information) by training on silver summaries generated by GPT-3.5. However, it requires inefficient autoregressive generation of the final context, and can eventually hallucinate facts not present in the input context. FilCo (Wang et al., 2023) similarly proposes to generate contexts autoregressively but is trained on extractive targets, i.e., one sentence from the context selected by one of three criteria. The drawbacks are again inefficiency and the simplified assumption of one relevant sentence per context. A recent approach, COMPACT (Yoon et al., 2024), also proposes to generate filtered contexts autoregressively – hence inefficiently – and introduces an iterative approach for gradually updating the relevant context after processing a new portion of retrieved passages. In contrast to all listed efforts, `Provence` dynamically detects the amount of relevant information in the context – from zero to all sentences – in an extractive and efficient way. Furthermore, we propose a novel approach of integrating context pruning into a reranker.

Concurrently to our work, DSLR (Hwang et al., 2024) performs extractive sentence-level pruning, by encoding sentences one-by-one, together with the query, using existing rerankers. Similarly to `Provence`, DSLR keeps sentences with scores higher than a threshold and preserves the original order of sentences. However, in contrast to Provence, `DSLR` processes sentences in a context independently, which could lead to misunderstanding of coreference between sentences and consequent pruning errors.

An orthogonal line of work proposes extractive *token-level* pruners. LLMLingua (Jiang et al., 2023) and Selective Context (Li et al., 2023) use LLMs to remove tokens with high generation probabilities, independently of the query. LLMLingua2 (Pan et al., 2024) is a small BERT-based model finetuned to eliminate redundant tokens, also independently of the query. LongLLMLingua (Jiang et al., 2024) proposes query-dependent LLM-based token pruning based on contrastive perplexity. Listed approaches remove tokens in a way that it does not break context understanding for the LLM – hence they are not capable of removing semantic parts of the context. LLMLingua models also have many hyperparameters in the interface which are hard to tune in practice. These approaches can however also be combined with sentence-level pruning.

**Retrieval granularity.** Alternatively to context pruning, one can reformulate datastore content into atomic units, e.g., *propositions* as in Dense-X retrieval Chen et al. (2024c) or *decontextualized sentences* (Choi et al., 2021). Such preprocessing is expensive and can lead to some information loss.

**Passage filtering.** Another related – and orthogonal – line of works focuses on filtering entire passages if they are deemed irrelevant for a given question; such an approach can be straightforwardly combined with `Provence`. A simple method consists in introducing a threshold on the (re)ranking score. LongLLMLingua reranks passages based on the probability of a question given the passage. (Yoran et al., 2024) use natural language inference models to filter out passages that do not entail question-answer pairs, but report that this approach sometimes filters out relevant passages too.

**Improving context processing in LLMs.** While context pruners aim to remove context parts irrelevant to the user's query, another line of work aims to process contexts more efficiently and effectively in LLMs. Efficient context processing could be achieved through efficient attention implementations (Dao, 2024; Anagnostidis et al., 2023), KV cache compression (Nawrot et al., 2024), encoding retrieved passages in parallel (Zhu et al., 2024), or compressing contexts into one or more context embeddings (Chevalier et al., 2023; Ge et al., 2024; Rau et al., 2024b; Louis et al., 2025). Other works aim to make LLMs more robust, by exposing them to noisy contexts during training or finetuning (Izacard et al., 2022; Lin et al., 2024). All such approaches usually require LLM adaptation which may complicate application to an arbitrary picked LLM.

## 3 PROVENCE

The high-level overview of our proposed approach is illustrated in Figure 1. Our first contribution is to pose the context pruning problem as a sequence labeling task. We fine-tune a DeBERTa model to encode the query–context pair and output binary masks which are used to filter out irrelevant context parts. The labels for training are generated by LLama-3-8B-Instruct (AI@Meta, 2024); we call them *silver labels* since they are generated automatically. Such an approach solves several limitations of existing context pruners: *(1)* by construction, the model is able to deal with varying noise in contexts and select an appropriate pruning ratio; *(2)* queries are encoded together with context sentences (cross-encoding), providing richer representations – compared for instance to extractive RECOMP which encodes query and context sentences independently; *(3)* using a lightweight encoder makes our approach more efficient than LLM-based or abstractive methods.

Our second contribution consists in unifying reranking and context pruning – instead of considering these steps as distinct in the RAG pipeline. In `Provence`, reranking and pruning can be done *in a single forward step*, thus eliminating the computational overhead due to context pruning – making `Provence` almost "free".

**Training data.** Our approach requires a set of training questions and a retrieval datastore. Specifically, we rely on the train set of the MS MARCO document ranking collection which includes $370k$ queries (Nguyen et al., 2016). The MS MARCO collection is a domain-diverse datastore of $3.2M$ documents crawled from the Web – which is required for the final model's robustness to various domains – and is often used to train retrievers and rerankers. In ablations, we also consider the train set of Natural Questions which contains $87k$ queries Kwiatkowski et al., 2019).

**Data processing.** We create a retrieval datastore by splitting MS MARCO documents into passages consisting of $N$ consecutive sentences – $N$ being a random integer $\in 1..10$. This is to enable the pruner's robustness to variable retrieved context lengths. We also prepend page titles to each passage. For each question, we retrieve top-5 relevant passages using a strong retrieval pipeline (Rau et al., 2024a) consisting of a SPLADE-v3 retriever (Lassance et al., 2024) and a DeBERTa-v3

reranker (Lassance & Clinchant, 2023). The resulting set of retrieved passages is naturally diverse w.r.t. relevance or irrelevance to the question, due to imperfections in retrieval.

**Silver labels generation.** Given a question and a retrieved passage (context), we split the passage into sentences[3] and prompt Llama-3-8B-Instruct to select sentences relevant to the given question. One approach would be to use a straightforward prompt such as *"Output indexes of sentences relevant to the given question"*. However, we decided to utilize the strong LLMs' capabilities of actually *answering* questions while *citing* relevant context sentences. We therefore instruct the LLM to answer the given question using *only* information provided in the given context, and output *"No answer"* in case no relevant information is provided. We also specify the easy-to-parse citation format [i] and number sentences with the same marker in the context. Our prompt can be found in Appendix – Table 6; we use greedy decoding and parse cited sentences using regular expressions. We also compare different prompting strategies in the ablation study.

We found that Llama-3-8B is well capable of answering only based on a given context in most cases and of outputting a citation $\sim 90\%$ of the time. We filter out cases when no citations are produced and *"No answer"* is not present in the LLM's output, as these are the cases when the context actually contains relevant information but the LLM "forgot" to cite it. The final labels distribution (number of selected sentences per context, their positions) is shown in Appendix – Figure 5.

**Training of Provence.** Our context pruner receives as input the concatenation of a question and a retrieved context, and outputs *per-token binary labels* denoting whether each token (defined by the pretrained model's tokenizer) should be included in the selected context. In Section 4.4 (Ablations), we also consider an approach where a special token is inserted at the beginning of each sentence, and labels are predicted per-sentence based on the representations of those tokens. We train `Provence` as a binary per-token classifier with ground truth labels coming from the silver data labeling, and the model can be used as a standalone pruner, i.e., `retrieve >> rerank >> Provence (standalone) >> generate`.

**Unifying compression and reranking.** We note that cross-encoder rerankers (Nogueira & Cho, 2020) share both the same architecture and inputs (pairs of question–passages) as `Provence`. Additionally, the task of context pruning (selecting parts of contexts that are useful for generating the answer to the question) intrinsically bears similarity with re-ranking (estimating the relevance of a context w.r.t. the question) – and we hypothesize the possibility of knowledge transfer between these two related tasks. We therefore propose to *unify* both approaches in a single model, with two different task heads. More specifically, the reranking head outputs a scalar prediction for the `BOS` token while the pruning head outputs per-token predictions for the passage tokens, as illustrated in Figure 1. To ease training, we propose to further fine-tune a pretrained reranker on our labeling objective, while adding a ranking "regularizer" to preserve initial reranking capabilities. The regularizer is a Mean Squared Error loss on the reranking scores from the initial reranker. This can be viewed as a straightforward pointwise score distillation process, where the initial model serves as the teacher – a method that has demonstrated great effectiveness in Information Retrieval Hofstätter et al. (2021). The final loss function is as follows:

$$\mathcal{L} = \sum_{n=1}^{N} \left\{ \sum_{k=1}^{L_n} \log P(y_{n,k}|z_{n,k}) + \lambda(s_n - z_{n,0})^2 \right\} \qquad z_n = \text{Provence}(x_n) \qquad (1)$$

where $N$ is the number of datapoints (query–passage pairs), $x_n$ is a sequence of $L_n + 1$ input tokens (concatenated query, passage and `BOS` at the 0-$th$ position), $z_n$ is a sequence of $L_n + 1$ predictions output by the model, $y_n$ is a sequence of $L_n$ target binary labels for context pruning, $s_n$ is the teacher score (initial reranker), $z_{n,0}$ is the ranking score predicted from the `BOS` representation.

In the case of the unified model, re-ranking and context pruning need a single forward step from the encoder, i.e., `retrieve >> Provence (w/ re-ranking) >> generate` – making context pruning almost free in terms of execution time.

**Inference with Provence.** At inference, we feed a concatenation of a question and a retrieved passage through `Provence`, which outputs probabilities of including each token in the final context, as well as the passage score in the case of the unified model. We simply use a threshold $T$ to binarize

---

[3]using the `nltk.sent_tokenize` function: `https://www.nltk.org/api/nltk.tokenize.sent_tokenize.html`

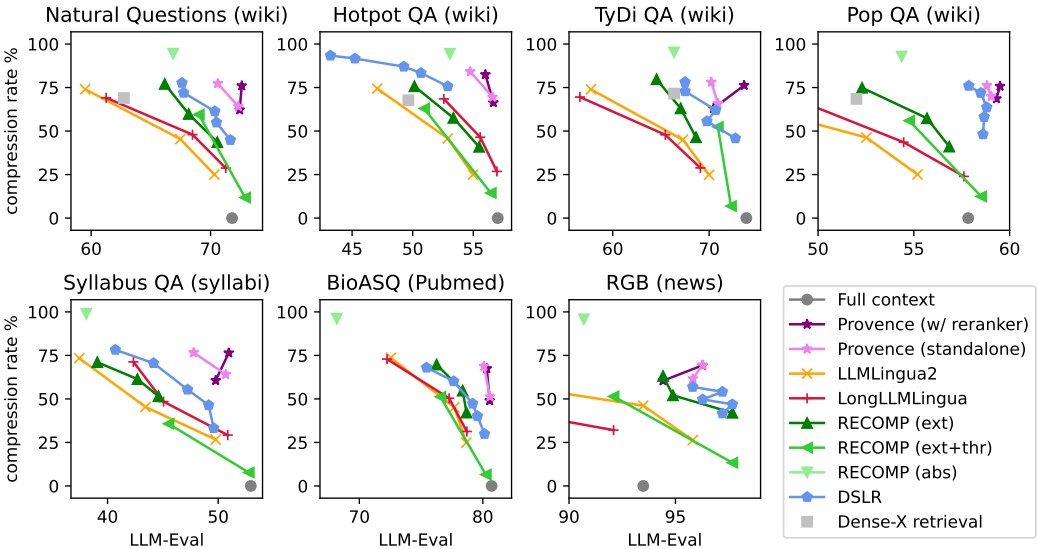

Figure 2: Main results for various QA domains, comparing `Provence` and baseline models. *Generator*: LLama-2-7B, *retriever*: SPLADE-v3, *reranker*: DeBERTa-v3 (or `Provence` in the unified setting). Plot titles denote "Dataset name (datastore type)". $x$-axis denotes QA performance evaluated with LLM-as-a-judge; $y$-axis denotes the context compression ratio. For both metrics, the higher the better: the best model would be closest to the top right corner. Numerical scores are presented in App. Tables 8–9. Main conclusion: `Provence` consistently lies on the Pareto front.

the token probabilites (keep or not) – which has a direct effect on the compression rate. As shown in the experiments section, the choice of a threshold is generally transferable across various datasets, making the model flexible to be used out-of-the box in various QA applications[4].

We note that our model outputs token-level predictions despite the sentence-level labeling task. We found that probabilities of including tokens into the final context are naturally clustered on the sentence level – see example in Appendix Figure 6 – due to the sentence-level targets used in training. However, in rare cases we could still have partial sentences being selected. To avoid this phenomenon, we apply a "sentence rounding" procedure: for each sentence, we check the ratio of kept tokens (predicted label= 1), and select the entire sentence only if it is higher than $0.5$.

## 4 EXPERIMENTS

### 4.1 EXPERIMENTAL SETUP

**Provence training details.** We train `Provence` on the MS Marco data processed as described in Section 3, using PyTorch (Paszke et al., 2019) and HuggingFace transformers (Wolf et al., 2020). We use DeBERTa-v3 (He et al., 2021a) as our pretrained model for training the standalone `Provence`. For the unified approach, we start training from an already trained cross-encoder, also based on DeBERTa-v3 (Lassance & Clinchant, 2023). Note that in the latter, we initialize the ranking head from its fine-tuned version, and train the separate pruning head from scratch.

After preliminary experiments, we set the learning rate to $3 \times 10^{-6}$, the batch size to 48 and train models for one epoch. For joint training, there is a slight trade-off between pruning and reranking. We set the reranking regularization coefficient $\lambda$ to $0.05$, chosen as the minimal value that does not substantially degrade reranking performance on the MS MARCO development set.

**Evaluation datasets.** We test `Provence` on a diverse set of QA datasets. First, we consider commonly used datasets relying on Wikipedia datastore: Natural Questions (Kwiatkowski et al., 2019),

---

[4]Note that tuning the threshold per dataset could of course further improve results.

Table 2: Time/MFLOPS required for context pruning, 50 samples, with batch size set 1 (1 sample consists of a query and top-5 retrieved documents).

| Pruner | Time ($s$) | FLOPS |
|---|---|---|
| LongLLMLingua, rate=0.5 | 194 | 5.400e+16 |
| LLMLingua2, rate=0.5 | 38 | 9.822e+15 |
| RECOMP extr., top=2 | 12 | 1.037e+15 |
| RECOMP abstr. | 499 | 5.357e+14 |
| DSLR, th=0.5 | 26 | 4.391e+15 |
| `Provence (standalone)` | 25 | 7.901e+14 |

Table 3: Speed up in generation due to compression (`Provence`, 49% compression). Batch sizes 1 or 256.

| Generator | $bs$ 1 | $bs$ 256 |
|---|---|---|
| LLama-2-7B | ×1.2 | ×2 |
| LLama-2-13B | ×1.4 | ×2 |
| SOLAR-10.7B | ×1.4 | ×1.9 |

TyDi QA (Clark et al., 2020), PopQA (Mallen et al., 2023b) (all three datasets include single-hop questions), and HotpotQA (Yang et al., 2018) (multi-hop questions). Second, we consider datasets with datastores from various domains: BioASQ (Nentidis et al., 2023) (biomedical questions with Pubmed as a datastore), SyllabusQA (Fernandez et al., 2024) (questions about educational course logistics, with courses syllabus as a datastore); and RGB Chen et al. (2024b) (questions about news with Google-searched news as contexts). Further details can be found in Appendix B.

**Evaluation settings.** We conduct experiments using BERGEN (Rau et al., 2024a), a benchmarking library for RAG, using the recommended experimental setting. For each query, we retrieve top-5 relevant passages using a strong and robust retrieval pipeline: SPLADE-v3 >> DeBERTa-v3 reranker (except for RGB, for which Google-searched passages are already provided). We then pass queries prepended with relevant document (full length or pruned) into LLama-2-7B-chat (Touvron et al., 2023)[5] to generate answers; other retrieval-generator settings are reported in Appendix. Each evaluation dataset comes with short keyword answers, which we use to evaluate responses using LLM-based evaluation (LLMeval in Rau et al., 2024a); match-based metrics are also reported in Appendix. We additionally measure compression as a portion of the context which was pruned out.

We compare `Provence` to publicly available context pruning models listed in Table 1, except LLMLingua and Selective Context which were shown to underperform LLMLingua2 (Pan et al., 2024). For all context pruners (except abstractive RECOMP for which it is not available), we enforce the selection of the first (title) passage sentence, to avoid ambiguity in understanding the context by the generator. For extractive RECOMP, we use the model trained on NQ, consider using top-1/2/3 sentences, and prepend the passage title to each sentence. For the LLMLingua family, we vary the compression rate in $\{0.25, 0.5, 0.75\}$ and use code provided on the official repository[6]. We use the XLM-RoBERTa model for LLMLingua2. For `Provence`, we use $T = 0.1$ and $T = 0.5$. We also compare our method to DSLR based on the same reranker as ours, i.e, DeBERTa-v3.

## 4.2 MAIN RESULTS

Context pruners are often only tested on limited domain data, e.g., with Wikipedia datastore, and an important aspect of our work is evaluating context pruning on a series of QA domains. Figure 2 reports the trade-off between compression (efficiency) and LLM-evaluated performance (quality), for various QA datasets and context pruning methods. We choose to report a figure per dataset to better assess the **Pareto front** of existing solutions, rather than comparing methods with different compression rates in the same table. Figure 7 in Appendix further reports similar results with match-based metric, and Appendix Tables 11–13 show examples of context pruning with various methods.

First, we observe that `Provence` achieves **the highest performance** across pruning methods, for similar compression ratios. Second, it is noteworthy that `Provence` outperforms methods requiring more computations such as LLMLingua models, showing that efficiency is not traded for effectiveness. Furthermore, `Provence` is the only method capable of achieving high compression levels without (or with negligible) performance drops, on all datasets. Moreover, for some datasets, e.g., PopQA, pruning with `Provence` leads to performance improvements due to noise filtering.

---

[5] For main experiments, we chose a "weaker" generator which relies more on contexts, to create a more challenging setting for context pruners; results with stronger generators are reported in Appendix – Figure 8.

[6] https://github.com/microsoft/LLMLingua

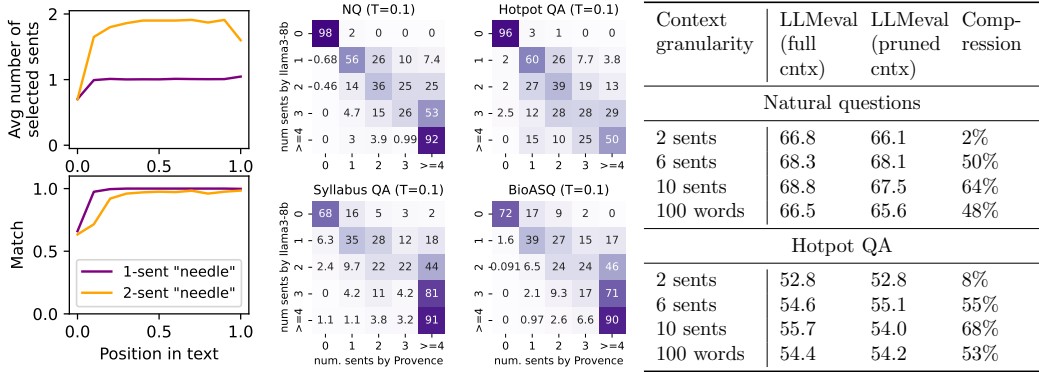

Figure 3: Analyses. (*Left*) Needle-in-the-haystack test allowing the control of the position of the ground truth sentence(s) in the context. (*Middle*) Comparison of the number of selected sentences by the silver predictor (LLaMA-3-8B-Instruct) and `Provence`. Heatmaps are normalized by rows: a cell in position $(i, j)$ indicates which percentage of contexts that were pruned into $i$ sentences by the silver predictor, were pruned into $j$ sentences by `Provence`. (*Right*) Testing `Provence` in settings with different context lengths. All experiments are done with unified `Provence`, $T = 0.1$.

**The effect of threshold.** An important aspect in the out-of-the-box applicability of context pruners is how much effort is needed to select the suitable values of hyperparameters. For `Provence`, it only consists in setting the pruning threshold $T$. In Figure 2 (for which $T = 0.1$ and $T = 0.5$), we observe that `Provence` pruning ratio automatically varies from 50% to 80%, depending on the dataset, which demonstrates that the same values for $T$ work well for all considered domains – making `Provence` robust to the choice of hyperparameters. If necessary, users can still tune it further for their datasets and/or needs. We note that some models specify the desired compression ratio as a hyperparameter, e.g., LLMLingua models or extractive RECOMP (through top-$N$ sentences). While it may seem convenient to estimate inference cost, the "optimal" compression ratio (without losing performance) is specific to each particular question-context pair. Thus, using a threshold as a hyperparameter is more appropriate for this task. We also experimented with specifying a threshold in extractive RECOMP (shown on the same plot) and found that it often leads to lower performance (compared to top-$N$). The reason is that different queries have different ranges of similarity scores.

**Efficiency.** We compare `Provence` with other pruning methods in terms of efficiency. Table 2 reports compression time and FLOPS[7] required by different pruning methods. As expected, LongLLMLingua (based on LLama-2-7B-chat) is much slower compared to other pruners based on smaller modles. RECOMP abstr. requires less FLOPS compared to `Provence`, but its autoregressive nature makes it very slow in practice. Note that in the case of the unified `Provence` model, pruning is almost free – as it's part of the re-ranking step. Table 3 reports speed-up gains due to compression with `Provence` model ($\sim$ 50% compression rate). All runs were performed on single Tesla V100-SXM2-32GB GPU with vllm Kwon et al. (2023). With large batch sizes, we systematically observe $2\times$ speed-ups at inference, while smaller batch sizes lead to lower gains (especially for smaller models). We assume this is mostly due to the CPU/GPU communication bottleneck, which masks inference gains due to compression.

## 4.3 ANALYSIS

This section reports a more fine-grained evaluation to better understand `Provence` properties.

**Robustness to the position of relevant information in the context.** We design a needle-in-the-haystack experiment which allows us to check the performance of `Provence` on a simple toy example and to evaluate its robustness w.r.t. the position of the relevant information in the input

---

[7]We use `https://github.com/cli99/flops-profiler` to report FLOPS required by each pruner. We note that FLOPS do not always align with real inference time.

Table 4: Effectiveness of reranking top-50 documents retrieved by SPLADE-v3. DeBERTa-v3 is the "baseline" (initialization point for `Provence`, which we aim to preserve performance). We report the R@5 on two RAG datasets (NQ and HotpotQA), MRR@10 on MS MARCO passages (dev set), nDCG@10 on TREC DL'19 (Craswell et al., 2020), and mean nDCG@10 on the 13 open datasets from the BEIR benchmark (Thakur et al., 2021) – Table 7 in Appendix reports the full results.

| Model | Dataset | | | | |
|---|---|---|---|---|---|
| | NQ | HotpotQA | MS | TREC19 | BEIR |
| DeBERTa-v3 | 83.0 | 70.4 | 40.5 | 77.4 | 55.4 |
| Provence | 84.4 | 70.5 | 40.6 | 77.2 | 55.9 |
| Provence (NQ) | 84.5 | 70.3 | 40.2 | 77.5 | 55.1 |

context. We write 5 questions and answers[8], and insert answers ("needles") at random positions between sentences, in a subset of 100 passages sampled from the Wikipedia datastore. Ideally, `Provence` should only select the "needle" sentences and filter out all other sentences in contexts. We plot the number of selected sentences and percentage of cases when the pruned context contains the "needle" (Figure 3, (*Left*)). We consider two settings: with 1- and 2-sentence "needles". We observe that `Provence` correctly selects "needle" sentence(s) in most cases, except at leftmost and rightmost positions.[9] In most cases `Provence` does not select any irrelevant sentences. The results are similar for both simpler (1-sentence) and harder (2-sentence) "needles" showing `Provence`'s flexibility in detecting the number of relevant sentences, discussed below in more details.

**Adaptability to the variable number of relevant sentences.** To evaluate the capability of `Provence` to dynamically detect the number of relevant sentences in the context, we compare the number of sentences $L$ selected by `Provence` and by a silver oracle. A silver oracle is easy to construct for $L = 0$, by pairing questions with randomly sampled contexts. For $L \geqslant 1$, we use the labeling produced by Llama-3-8B-Instruct. Figure 3 (*Middle*) shows that the number of relevant sentences detected by `Provence` is close to the silver oracle value in most cases, for all considered datasets. In contrast, extractive RECOMP would always select a prespecified number of sentences.

**Robustness w.r.t. context granularity.** Figure 3 (*Right*) shows `Provence` performance for two datasets, with Wikipedia datastores made of contexts of various granularity. Here, each considered datastore is produced by splitting Wikipedia pages into chunks of $N$ sentences, $N \in \{2, 6, 10\}$, or 100 words, and prepending the page title to each chunk. `Provence` shows high performance in all cases – the performance with pruned contexts being close to the performance obtained using original contexts. As could be expected, the compression ratio is higher for longer contexts.

**Reranking effectiveness.** Table 4 compares **reranking** performance between our reranking baseline and unified `Provence` – whose training starts from the former. We can see that our joint training procedure (on both pruning and ranking tasks) makes it possible to learn a context pruner that preserves initial reranking capabilities. We further include as a comparison point results from a model trained in similar conditions on NQ. Overall, results are similar – further highlighting the robustness of `Provence` w.r.t. training data. We further discuss such aspects in Section 4.4 (Ablations).

**Applicability in different settings.** Figure 8 (App.) demonstrates the applicability of `Provence` in variable retrieval-generator settings – achieving similar results as the ones reported in Figure 2.

## 4.4 ABLATIONS

In this Section we analyze various design choices made in `Provence` development, to provide insights into training context pruners for future works (results shown in Figure 4). All models in this section are standalone context pruners, trained with the same amount of parameter updates.

---

[8]Example: "Which library was used in the experiments?", answer: "Experiments were conducted using the Bergen library". Example reformulation into a 2-sentence answer: "Experiments were conducted using a library. Its name is Bergen."

[9]The reason for the drops in the left-most and right-most positions is that training data has little examples of the corresponding types of relevant sentences, see e.g., statistics for the rightmost position in the App. Figure 5, *(Right)*. We leave further improving processing of these positions to future work.

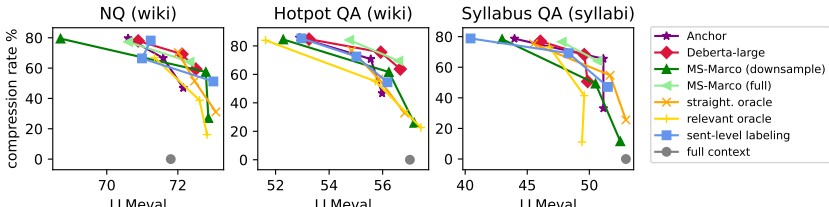

Figure 4: Ablation results. All models are single-component modifications of the anchor model, which is a base-size model, trained on NQ, with the answer oracle and token-level labeling. Numeric scores / match-based results are presented in Appendix Table 10 / Figure 11 correspondingly.

**Model size.** We first observe that DeBERTa-large slighly increases the compression rate – when comprared to DeBERTa-base. All other ablations are tuned from a DeBERTa-base model, for efficiency reasons. Note that the final `Provence` is trained from a DeBERTa-large model (or its equivalent reranker).

**Data mixtures.** We compare training on NQ ($87k$ queries), MS MARCO downsampled to the same size, and full MS MARCO ($370k$ queries). Despite the observation that using the MS MARCO *type of data* leads to lower results than NQ – with equal number of queries – we also find that using larger data (i.e., full MS MARCO) improves results. Our final models are trained on the full MS MARCO – further ablations are conducted on the NQ data, for efficiency reasons.

**Labeling strategies.** As described in Section 3, we can train the pruner either to perform token-level labeling (with sentence rounding at inference) or to perform sentence-level labeling. In the former case sentence representations are richer but the model also needs to learn to output similar predictions for tokens inside one sentence. In the latter case sentence content must be represented in a single embedding which may limit representation expressivity. In practice we observe close performance, with the token-level strategy slightly outperforming the sentence-level one some datasets. In all other experiments we use the token-level strategy.

**Oracle prompts.** We compare three options for prompting an oracle LLM to generate silver labeling: *(1) answer oracle*: asking to answer the given question from the given context, citing corresponding sentences; *(2) relevance oracle*: asking to list any *relevant* information in the context to the question, citing corresponding sentences; *(3) straightforward oracle*: asking to output indexes of sentences which answer the given question. We found that the behavior of the *straightforward oracle* varies on different prompts, while the use of the *answer oracle* makes answers more consistent. The motivation for the *relevance oracle* is that often contexts contain distantly relevant information to the query and it could be reasonable to select the corresponding sentences. Comparing the listed prompts, we observe that the *relevance oracle* underperforms the *answer oracle*, and the *straightforward oracle* performs similarly or slightly lower than the *answer oracle*.

**Unification with reranker.** In Figure 2 we compare `Provence` trained as a standalone model and as a model unified with reranker, and find that both strategies lead to similar results – although the former relies on two separate inference steps (re-ranking and pruning) in a RAG pipeline.

## 5 CONCLUSION

In this work, we present `Provence`, a robust, adaptable, and efficient context pruner for Question Answering – either unified in a single model with reranking capabilities or available as a lightweight standalone model. In contrast to previous extractive approaches, `Provence` dynamically detects the needed pruning ratio for a given context and can be used out-of-the-box for various QA domains. In extensive experiments, we demonstrate that `Provence` prunes contexts with negligible to no drops in performance and in some cases even brings performance improvement due to removing context noise. We also show `Provence` capabilities in correctly detecting the number of relevant sentences in contexts, located at any position, and with contexts of various lengths. Finally, the ablation study highlights the importance of using a large training data and the appropriate prompt in the silver oracle.

**Limitations.** Despite `Provence` being ready to use in various settings, demonstrated in the paper, it is focusing only on QA applications, with a single passage processed at a time, and is trained on English-only data. Future work could consider extending it to other tasks, multi-passage contexts, and languages beyond English.

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

## A  CODE AND EXPERIMENTS DATA

We release our code at `https://github.com/naver/bergen/tree/main/scripts/provence`. In this page, we also make available a zip archive with experimental results for the main Figure 2, e.g. it includes model generations and scores computed with various evaluation metrics (LLMEval, Match, Recall etc).

## B  DATA

**Evaluation datasets.**  We consider the following datasets:

- Datasets with Wikipedia as a datastore:
  - Natural Questions (Kwiatkowski et al., 2019). We use a test set of $2.8k$ questions, distributed as a part of the KILT collection (`https://huggingface.co/datasets/facebook/kilt_tasks`);
  - HotpotQA (Yang et al., 2018). We use a test set of $5.6k$ questions, distributed as a part of the KILT collection (`https://huggingface.co/datasets/facebook/kilt_tasks`);
  - PopQA (Mallen et al., 2023b). We use a test set of $14k$ questions distributed by the dataset authors.
- Datasets with individual datastores:
  - BioASQ (Nentidis et al., 2023). We use a version of the dataset provided by (Hsia et al., 2024), with $3.8k$ queries. We only use queries from categories "yes/no", "factoid", and "list".
  - Syllabus QA (Fernandez et al., 2024). We use the test set of $1.1k$ questions distributed by the authors;
  - RGB (Chen et al., 2024b). We use the test set of 200 questions distributed by the authors.

All datasets provide short answers (keywords) for each query, which we use to evaluate both match-based metrics such as Recall and LLM-based metrics Rau et al. (2024a)[10].

**Datastores.**  For training `Provence`, we use the MS MARCO document collection (Craswell et al., 2021). We split each document into overlapping chunks of $N$ sentences, where $N$ is random in $\in 1..10$ – with a higher probability for longer contexts – to train `Provence` on various context lengths. Each chunk is prepended with a page title. The resulting datastore contains $34M$ passages. We also process the Wikipedia datastore in a similar fashion, for ablation experiments. We download a 2024 Wikipedia dump and process it using scripts provided by Pyserini (Lin et al., 2021)[11]. We also prepare versions of this Wikipedia datastore with passages of $N$ sentences with overlaps of $N/2$ sentences, for testing `Provence` robustness to various context lengths.

All other evaluations on Wikipedia-based datasets – including main evaluations – are conducted on the Wikipedia datastore provided at `https://huggingface.co/datasets/castorini/odqa-wiki-corpora`. We use a version with passages of 6 sentences with a 3-sentence overlap – making $9M$ passages in total.

For Pubmed, we use the version of the dataset provided by (Hsia et al., 2024) at `https://huggingface.co/datasets/jenhsia/ragged`. It consists of $58M$ passages, extracted from Pubmed abstracts. Each passage (chunk) is prepended with the page's title.

For SyllabusQA, we split each syllabus (provided by the authors) into passages of 100 words. For RGB, Google-retrieved contexts are provided by the authors; for each query we compose a retrieved context from 3 relevant and 2 irrelevant documents.

---

[10]Using SOLAR-10.7B (Kim et al., 2023).
[11]At `https://github.com/castorini/pyserini/blob/master/docs/experiments-wiki-corpora.md`.

## C  MODELS

We list in Table 5 all the main models used to conduct experiments for `Provence`.

| Model | Checkpoint |
|---|---|
| SPLADE-v3 | `naver/splade-v3` |
| RetroMAE | `Shitao/RetroMAE_MSMARCO_distill` |
| DeBERTa-v3 | `microsoft/deberta-large` |
| DeBERTa-v3 (RR) | `naver/trecdl22-crossencoder-debertav3` |
| BGE-M3 | `BAAI/bge-reranker-v2-m3` |
| LLama-2-7B-chat | `meta-llama/Llama-2-7b-chat-hf` |
| LLaMA-3-8B-Instruct | `meta-llama/Meta-Llama-3-8B-Instruct` |
| Mistral-7B-instruct | `mistralai/Mistral-7B-Instruct-v0.2` |
| SOLAR-10.7B-Instruct-v1.0 | `upstage/SOLAR-10.7B-Instruct-v1.0` |

Table 5: List of all the models used in the experiments with their corresponding HuggingFace checkpoints.

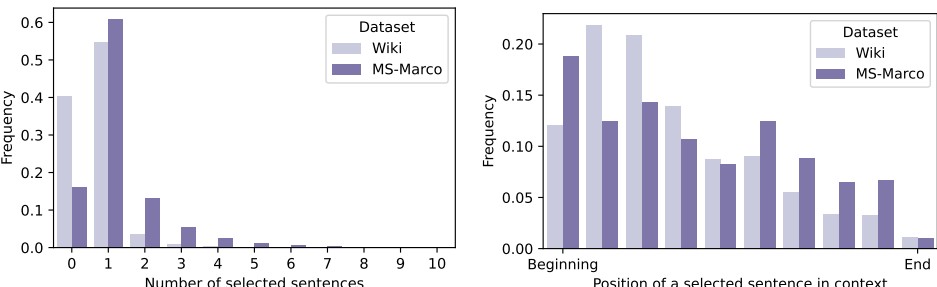

Figure 5: Statistics of the silver contexts labeled by LLaMA-3-8B-Instruct. *(Left)* the distribution of the number of sentences in silver contexts. *(Right)* the distribution of the position of the selected sentences in contexts.

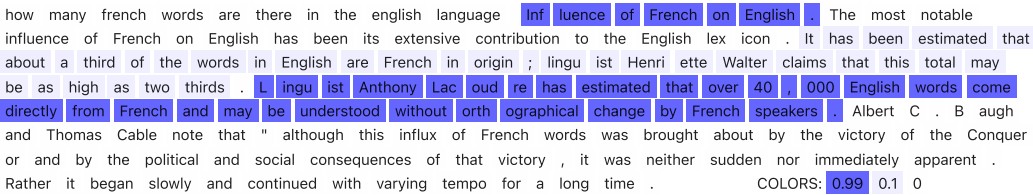

Figure 6: Example visualization of per-token probabilities of being selected in the final context.

Table 6: Prompt used for generating silver labeling with LLaMA-3-8B-Instruct. The sentence citations in the response are parsed using regular expression.

---

Question: {question}
Context: [1] {sentence1} [2] {sentence2} [3] {sentence3} ...
Answer the Question, using ONLY information provided in the Context. If no useful information is provided, you MUST output "No answer". If some parts of the Context are used to answer, you MUST cite ALL the corresponding sentences. Use the symbols [ ] to indicate when a fact comes from a sentence in the context, e.g [0] for a fact from sentence 0. You should only answer the given question and should not provide any additional information.

---

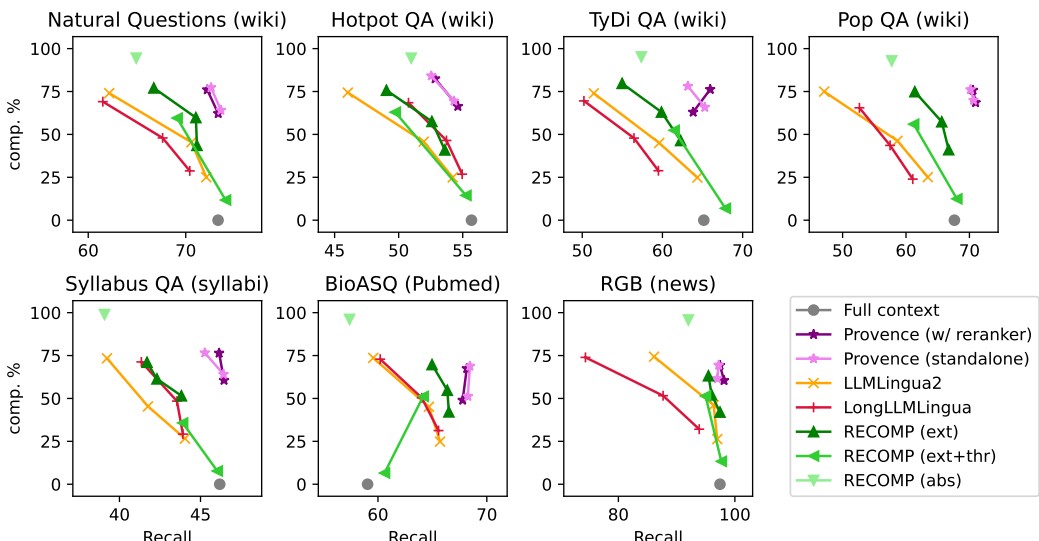

Figure 7: Main results for various QA domains, comparing `Provence` and baseline models, metric: Recall. *Generator*: LLama-2-7B, *retriever*: SPLADE-v3, *reranker*: DeBERTa-v3 (or `Provence` in the unified setting). Plot titles denote "Dataset name (datastore type)". $x$-axis denotes QA performance evaluated with Recall; $y$-axis denotes the context compression ratio. For both metrics, the higher the better: the best model would be closest to the top right corner.

Table 7: nDCG@10 on the 13 open BEIR datasets.

| Corpus | DeBERTav3 | Provence |
|---|---|---|
| TREC-COVID | 88.3 | 88.3 |
| NFCorpus | 37.5 | 37.8 |
| NQ | 66.7 | 66.5 |
| HotpotQA | 74.5 | 74.9 |
| FiQA-2018 | 47.8 | 47.6 |
| ArguAna | 29.8 | 33.2 |
| Touché-2020 | 33.5 | 33.4 |
| Quora | 84.8 | 85.4 |
| DBPedia | 48.9 | 49.2 |
| SCIDOCS | 19.2 | 19.6 |
| FEVER | 86.6 | 87.9 |
| Climate-FEVER | 27.4 | 28.1 |
| SciFact | 75.8 | 75.3 |
| average | 55.4 | 55.9 |

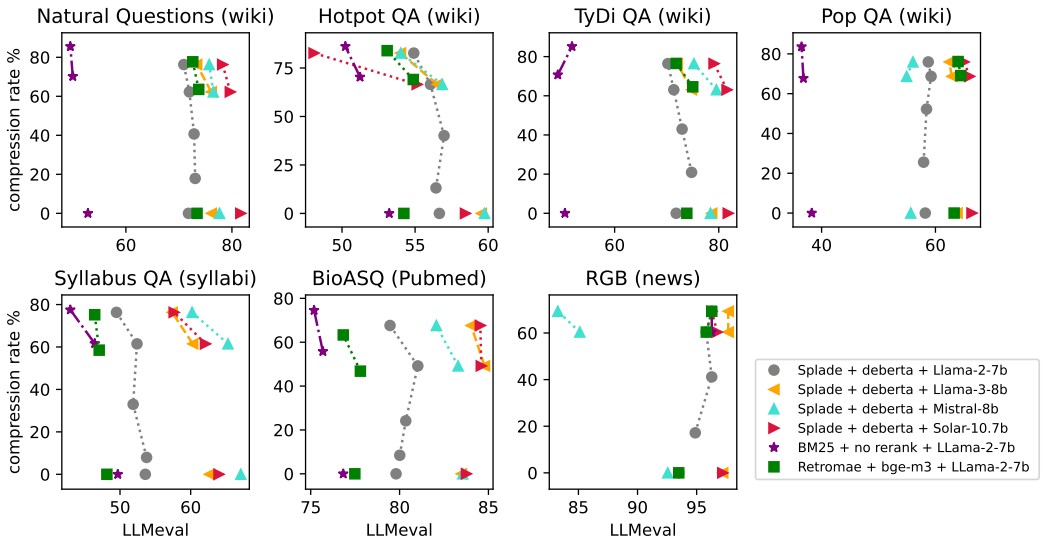

Figure 8: Testing `Provence` in various RAG settings (retrieval, re-ranking, generator).

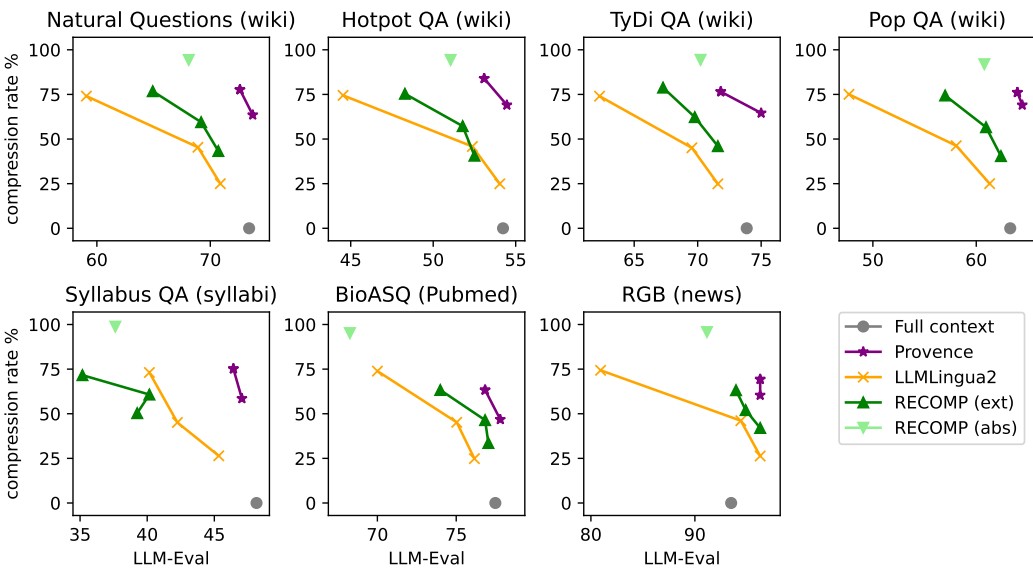

Figure 9: Comparing `Provence` to a subset of baselines with *retriever*: RetroMAE (Shitao et al., 2022), *reranker*: BGE-M3 (Chen et al., 2024a), *generator*:: LLama-2-7B-chat.

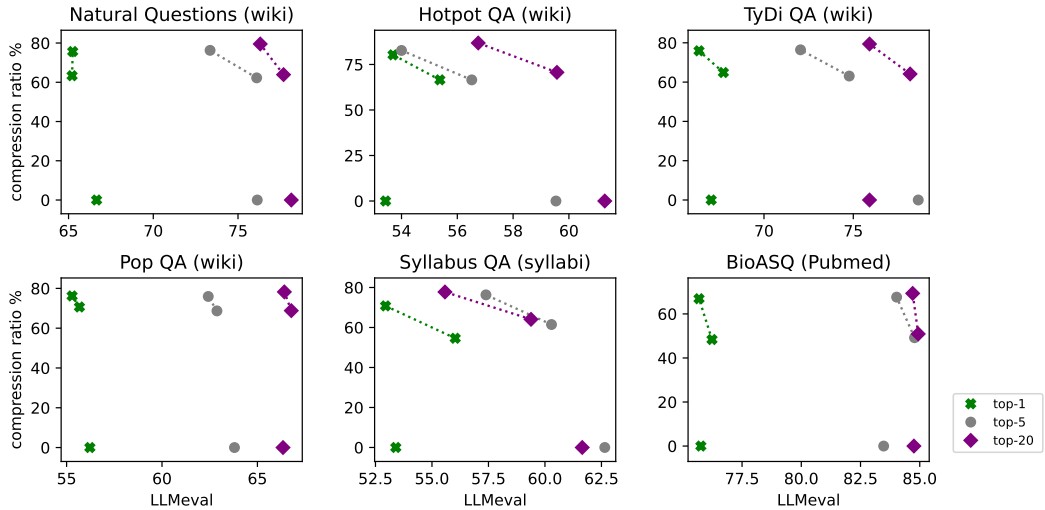

Figure 10: Testing `Provence` with different top-$k$ documents provided to the generator. The setting is the same as the one in Figure 2.

Table 8: Numerical scores corresponding to Figure 2 – NQ, Hotpot QA, Tydi QA, and Pop QA.

|  | NQ | | HotPot QA | | Tydi QA | | PopQA | |
|---|---|---|---|---|---|---|---|---|
|  | LLM-Eval | Comp. rate % | LLM-Eval | Comp. rate % | LLM-Eval | Comp. rate % | LLM-Eval | Comp. rate % |
| Full context | 71.8 | 0.0 | 57.0 | 0.0 | 73.9 | 0.0 | 57.8 | 0.0 |
| Provence (w/ reranker) | 72.4 | 62.2 | 56.7 | 66.4 | 70.5 | 63.0 | 59.3 | 68.6 |
|  | 72.6 | 76.0 | 56.0 | 82.4 | 73.6 | 76.2 | 59.5 | 75.8 |
| Provence (standalone) | 72.3 | 64.1 | 56.6 | 69.5 | 70.9 | 65.8 | 59.0 | 69.9 |
|  | 70.6 | 77.3 | 54.8 | 84.1 | 70.2 | 78.1 | 58.8 | 76.1 |
| LLMLingua2 | 59.5 | 74.0 | 47.1 | 74.4 | 57.7 | 73.9 | 42.9 | 75.0 |
|  | 67.5 | 45.4 | 52.9 | 45.8 | 67.3 | 45.0 | 52.5 | 46.3 |
|  | 70.3 | 25.0 | 55.0 | 24.9 | 70.0 | 24.8 | 55.2 | 25.1 |
| LongLLMLingua | 61.3 | 69.1 | 52.6 | 68.5 | 56.6 | 69.5 | 49.5 | 65.5 |
|  | 68.5 | 47.9 | 55.6 | 46.5 | 65.5 | 47.8 | 54.5 | 43.6 |
|  | 71.3 | 28.7 | 56.9 | 26.8 | 69.1 | 28.8 | 57.6 | 23.9 |
| RECOMP (ext) | 70.6 | 43.6 | 55.5 | 40.9 | 68.6 | 46.4 | 56.9 | 41.1 |
|  | 68.2 | 59.8 | 53.4 | 57.5 | 67.0 | 63.0 | 55.7 | 57.4 |
|  | 66.2 | 77.1 | 50.1 | 75.7 | 64.5 | 79.7 | 52.3 | 74.9 |
| RECOMP (ext+thr) | 69.0 | 59.5 | 50.9 | 62.9 | 70.9 | 52.4 | 54.8 | 56.0 |
|  | 72.9 | 11.8 | 56.4 | 14.4 | 72.3 | 6.9 | 58.5 | 12.4 |
| RECOMP (abs) | 66.9 | 94.5 | 53.1 | 94.4 | 66.4 | 95.2 | 54.4 | 92.8 |
| DSLR | 71.7 | 44.9 | 52.9 | 75.7 | 72.7 | 45.8 | 58.6 | 48.1 |
|  | 70.5 | 54.9 | 50.7 | 83.4 | 69.8 | 55.6 | 58.7 | 58.1 |
|  | 70.4 | 61.4 | 49.3 | 87.0 | 70.7 | 62.0 | 58.8 | 63.7 |
|  | 67.7 | 72.0 | 45.2 | 91.7 | 67.5 | 72.9 | 58.5 | 71.9 |
|  | 67.6 | 77.7 | 43.2 | 93.4 | 67.5 | 78.1 | 57.9 | 76.0 |
| Dense-X retrieval | 62.7 | 69.0 | 49.6 | 67.7 | 66.4 | 71.5 | 52.0 | 68.5 |

Table 9: Numerical scores corresponding to Figure 2: Syllabus QA, BioASQ, and RGB.

| | Syllabus QA | | BioASQ | | RGB | |
|---|---|---|---|---|---|---|
| | LLM-Eval | Comp. rate % | LLM-Eval | Comp. rate % | LLM-Eval | Comp. rate % |
| Full context | 52.9 | 0.0 | 80.7 | 0.0 | 93.5 | 0.0 |
| Provence (w/ reranker) | 49.8 | 60.6 | 80.6 | 49.0 | 94.4 | 60.5 |
| | 51.0 | 76.5 | 80.3 | 67.4 | 96.3 | 69.3 |
| Provence (standalone) | 50.7 | 64.1 | 80.6 | 51.3 | 95.8 | 61.6 |
| | 47.8 | 76.6 | 80.1 | 68.9 | 96.3 | 69.4 |
| LLMLingua2 | 37.4 | 73.4 | 72.6 | 73.6 | 78.6 | 74.3 |
| | 43.4 | 45.4 | 77.7 | 45.2 | 93.5 | 46.1 |
| | 49.8 | 26.6 | 78.7 | 24.8 | 95.8 | 26.3 |
| LongLLMLingua | 42.3 | 71.3 | 72.2 | 72.9 | 71.6 | 73.9 |
| | 45.1 | 48.5 | 77.3 | 50.4 | 83.3 | 51.6 |
| | 50.9 | 29.2 | 78.7 | 31.3 | 92.1 | 32.1 |
| RECOMP (ext) | 44.6 | 51.5 | 78.7 | 42.2 | 97.7 | 42.0 |
| | 42.7 | 61.4 | 78.4 | 54.8 | 94.9 | 52.1 |
| | 39.1 | 71.1 | 76.3 | 69.7 | 94.4 | 63.2 |
| RECOMP (ext+thr) | 45.5 | 35.7 | 76.6 | 51.2 | 92.1 | 51.4 |
| | 52.8 | 7.7 | 80.2 | 6.5 | 97.7 | 13.4 |
| RECOMP (abs) | 38.1 | 98.9 | 68.2 | 96.1 | 90.7 | 95.7 |
| DSLR | 49.6 | 33.2 | 80.1 | 29.9 | 97.2 | 41.6 |
| | 49.1 | 46.4 | 79.6 | 40.1 | 97.7 | 46.9 |
| | 47.2 | 55.4 | 79.2 | 47.3 | 96.3 | 49.7 |
| | 44.2 | 70.6 | 77.6 | 60.2 | 97.2 | 54.1 |
| | 40.7 | 78.2 | 75.4 | 68.0 | 95.8 | 56.9 |

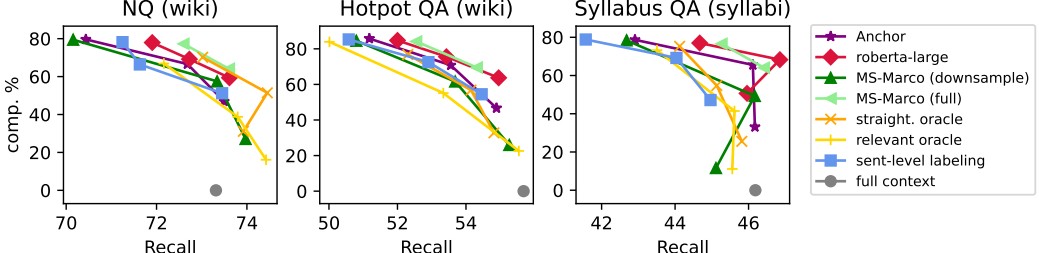

Figure 11: Ablation results with Recall (match-based metric).

Table 10: Numerical scores corresponding to Figure 4.

| | Thresh. | NQ | | HotPot QA | | Syllabus QA | |
|---|---|---|---|---|---|---|---|
| | | LLM-Eval | Comp. rate % | LLM-Eval | Comp. rate % | LLM-Eval | Comp. rate % |
| Anchor | 0.01 | 72.2 | 46.9 | 56.0 | 46.7 | 51.1 | 33.1 |
| | 0.1 | 71.6 | 66.6 | 55.6 | 70.8 | 51.1 | 65.5 |
| | 0.5 | 70.6 | 79.6 | 52.9 | 85.8 | 44.0 | 78.7 |
| Deberta-large | 0.01 | 72.5 | 59.4 | 56.7 | 63.7 | 49.9 | 50.7 |
| | 0.1 | 72.1 | 69.2 | 55.9 | 75.6 | 49.6 | 68.3 |
| | 0.5 | 70.9 | 78.0 | 53.2 | 84.7 | 46.1 | 77.0 |
| MS-Marco (downsample) | 0.01 | 72.9 | 27.1 | 57.2 | 25.9 | 52.5 | 11.6 |
| | 0.1 | 72.8 | 57.5 | 56.2 | 61.6 | 50.5 | 49.3 |
| | 0.5 | 68.7 | 79.4 | 52.3 | 84.5 | 43.0 | 78.4 |
| MS-Marco (full) | 0.1 | 72.3 | 64.1 | 56.6 | 69.5 | 50.7 | 64.1 |
| | 0.5 | 70.6 | 77.3 | 54.8 | 84.1 | 47.8 | 76.6 |
| straight. oracle | 0.01 | 73.1 | 31.1 | 56.8 | 32.8 | 52.9 | 25.6 |
| | 0.1 | 72.5 | 51.4 | 55.9 | 56.8 | 51.7 | 54.7 |
| | 0.5 | 72.0 | 70.2 | 54.8 | 76.8 | 45.5 | 75.3 |
| relevant oracle | 0.01 | 72.8 | 16.1 | 57.4 | 22.6 | 49.4 | 11.1 |
| | 0.1 | 72.6 | 38.8 | 55.7 | 55.1 | 49.6 | 41.4 |
| | 0.5 | 71.3 | 66.9 | 51.6 | 83.9 | 46.8 | 73.3 |
| sent-level labeling | 0.01 | 73.0 | 51.2 | 56.2 | 54.5 | 51.5 | 47.2 |
| | 0.1 | 71.0 | 66.4 | 55.0 | 72.5 | 48.3 | 69.2 |
| | 0.5 | 71.2 | 78.2 | 53.0 | 85.3 | 40.4 | 78.8 |
| full context | 0.01 | 71.8 | 0.0 | 57.0 | 0.0 | 52.9 | 0.0 |

Table 11: Example of context pruning with various approaches. `Provence` selects one sentence about the Shepard's pie and removes sentences about other similar dishes, which is RECOMP (ext) is not capable of by design. RECOMP (abs) correctly generates a summary; LongLLMLingua removes the part relevant to the Shepard's pie, and LLMLingua2 uniformly removes no-informative tokens.

| Question | what goes on the bottom of shepherd's pie |
|---|---|
| Original context | Shepherd's pie. History. In early cookery books, the dish was a means of using leftover roasted meat of any kind, and the pie dish was lined on the sides and bottom with mashed potato, as well as having a mashed potato crust on top. Variations and similar dishes. Other potato-topped pies include: The modern "Cumberland pie" is a version with either beef or lamb and a layer of breadcrumbs and cheese on top. In medieval times, and modern-day Cumbria, the pastry crust had a filling of meat with fruits and spices.. In Quebec, a variation on the cottage pie is called "Pâté chinois". It is made with ground beef on the bottom layer, canned corn in the middle, and mashed potato on top.. The "shepherdess pie" is a vegetarian version made without meat, or a vegan version made without meat and dairy.. In the Netherlands, a very similar dish called "philosopher's stew" () often adds ingredients like beans, apples, prunes, or apple sauce.. In Brazil, a dish called in refers to the fact that a manioc puree hides a layer of sun-dried meat. |
| `Provence` | Shepherd's pie. In early cookery books, the dish was a means of using leftover roasted meat of any kind, and the pie dish was lined on the sides and bottom with mashed potato, as well as having a mashed potato crust on top. |
| RECOMP (ext, top-3) | In early cookery books, the dish was a means of using leftover roasted meat of any kind, and the pie dish was lined on the sides and bottom with mashed potato, as well as having a mashed potato crust on top. The "shepherdess pie" is a vegetarian version made without meat, or a vegan version made without meat and dairy.. In the Netherlands, a very similar dish called "philosopher's stew" () often adds ingredients like beans, apples, prunes, or apple sauce.. |
| RECOMP (abs) | Shepherd's pie is a dish made with leftover roasted meat and mashed potato on the bottom, with variations including a version with beef or lamb and a layer of breadcrumbs and cheese on top. The dish has a history of being used as a means of using leftover roasted meat and has variations on the toppings. |
| LLMLingua2 (comp.50%) | Shepherd's pie History early cookery books dish leftover roasted meat lined mashed potato mashed potato crust top Variations similar dishes potato-topped pies include modern "Cumberland pie" beef or lamb breadcrumbs cheese medieval modern-day Cumbria pastry crust filling meat fruits spices Quebec variation cottage pie "Pâté ground beef bottom canned corn middle mashed potato top "shepherdess pie" vegetarian without meat vegan version without meat dairy Netherlands similar dish "philosopher's stew" adds ingredients beans apples prunes apple sauce Brazil dish manioc puree hides sun-dried meat |
| LongLLMLingua (comp. 50%). Processes all passages together and dynamically decides on the compression ratio of each passage. | Shepherd's pie. Other potato-topped pies include: The modern "Cumberland pie" is a version with either beef or lamb and a layer of breadcrumbs and cheese on top. In medieval times, and modern-day Cumbria, the pastry crust had a filling of meat with fruits and spices.. In Quebec, a variation on the cottage pie is called "Pâté chinois". It is made with ground beef on the bottom layer, canned corn in the middle, and mashed potato on top.. The "shepherdess pie" is a vegetarian version made without meat, or a vegan version made without meat and dairy.. In the Netherlands, a very similar dish called "philosopher's stew" () often adds ingredients like beans, apples, prunes, or apple sauce.. In Brazil, a dish called in refers to the fact that a manioc puree hides a layer of sun-dried meat. |

Table 12: Example of context pruning with various approaches. `Provence` correctly detects that the entire passage is relevant to the query, same as LongLLMLingua, while RECOMP (ext) is by design not capable of making such a decision.

| Question | where does the sweetness of fruit come from |
|---|---|
| Original context | Sweetness. A number of plant species produce glycosides that are sweet at concentrations much lower than sugar. The most well-known example is glycyrrhizin, the sweet component of licorice root, which is about 30 times sweeter than sucrose. Another commercially important example is stevioside, from the South American shrub "Stevia rebaudiana". It is roughly 250 times sweeter than sucrose. Another class of potent natural sweeteners are the sweet proteins such as thaumatin, found in the West African katemfe fruit. Hen egg lysozyme, an antibiotic protein found in chicken eggs, is also sweet. |
| `Provence` | Sweetness. A number of plant species produce glycosides that are sweet at concentrations much lower than sugar. The most well-known example is glycyrrhizin, the sweet component of licorice root, which is about 30 times sweeter than sucrose. Another commercially important example is stevioside, from the South American shrub "Stevia rebaudiana". It is roughly 250 times sweeter than sucrose. Another class of potent natural sweeteners are the sweet proteins such as thaumatin, found in the West African katemfe fruit. Hen egg lysozyme, an antibiotic protein found in chicken eggs, is also sweet. |
| RECOMP (ext, top-3) | It is roughly 250 times sweeter than sucrose. Another commercially important example is stevioside, from the South American shrub "Stevia rebaudiana". A number of plant species produce glycosides that are sweet at concentrations much lower than sugar. |
| RECOMP (abs) | [empty context] |
| LLMLingua2 (comp.50%) | Sweetness plant species produce glycosides sweet lower sugar glycyrrhizin sweet licorice root 30 times sweeter sucrose stevioside South American shrub "Stevia 250 times sweeter sucrose sweeteners sweet proteins thaumatin West African katemfe fruit Hen egg lysozyme antibiotic protein chicken eggs sweet |
| LongLLMLingua (comp. 50%) | Sweetness. A number of plant species produce glycosides that are sweet at concentrations much lower than sugar. The most well-known example is glycyrrhizin, the sweet component of licorice root, which is about 30 times sweeter than sucrose. Another commercially important example is stevioside, from the South American shrub "Stevia rebaudiana". It is roughly 250 times sweeter than sucrose. Another class of potent natural sweeteners are the sweet proteins such as thaumatin, found in the West African katemfe fruit. Hen egg lysozyme, an antibiotic protein found in chicken eggs, is also sweet. |

Table 13: Example of context pruning with various approaches. Provence selects one most relevant sentence, which is also ranked first by RECOMP (ext). RECOMP (abs) decides that no information is relevant to the query, while LongLLMLingua on the contrary keeps the entire input, dropping some punctuation marks. LLMLingua2 removes too many tokens which makes text hardly understandable.

| Question | what was the tower of london originally used for |
|---|---|
| Original context | Tower of London. In the 16th century, the Tower acquired an enduring reputation as a grim, forbidding prison. This had not always been the case. As a royal castle, it was used by the monarch to imprison people for various reasons, however these were usually high-status individuals for short periods rather than common citizenry as there were plenty of prisons elsewhere for such people. Contrary to the popular image of the Tower, prisoners were able to make their life easier by purchasing amenities such as better food or tapestries through the Lieutenant of the Tower. As holding prisoners was originally an incidental role of the Tower – as would have been the case for any castle – there was no purpose-built accommodation for prisoners until 1687 when a brick shed, a "Prison for Soldiers", was built to the north-west of the White Tower. The Tower's reputation for torture and imprisonment derives largely from 16th-century religious propagandists and 19th-century romanticists. |
| Provence | Tower of London. As a royal castle, it was used by the monarch to imprison people for various reasons, however these were usually high-status individuals for short periods rather than common citizenry as there were plenty of prisons elsewhere for such people. |
| RECOMP (ext, sorted top-3 sents) | As a royal castle, it was used by the monarch to imprison people for various reasons, however these were usually high-status individuals for short periods rather than common citizenry as there were plenty of prisons elsewhere for such people. This had not always been the case. The Tower's reputation for torture and imprisonment derives largely from 16th-century religious propagandists and 19th-century romanticists. |
| RECOMP (abs) | [empty context] |
| LLMLingua2 (comp.25%) | Tower London 16th century grim prison royal castle monarch high-status common citizenry prisoners amenities food Lieutenant Tower no-built accommodation until 1687 "Prison for north-west White Tower reputation torture imprisonment 16th-century propagandists 19th-century romanticists |
| LongLLMLingua (comp. 25%) | Tower of London In the 6th century, the acquired an enduring reputation as grim, forbidd prison. This had always been the case As a royal castle, it was by the to imprison people for various reasons however these were usually high-status individuals for short rather than common citizenry as there were plenty of prisons elsewhere for such people. Contrary popular of the Tower, prisoners were able to make their life easier purchasing amenities such better food or tapestries through Lieutenant of the Tower. holding prisoners was originally incident role of the– would have been the case for any – was purpose- accommodation for prisoners until 167 a "Prison for Sold", was to thewest of White Tower. The's reputation torture imprisonment derives largely from 6th- religious propagandists and 19th-century romanticists. |

