# OpenReview forum: "Provence: efficient and robust context pruning for retrieval-augmented generation"
_ICLR.cc/2025/Conference — ICLR 2025 Poster_

### Official Review · Reviewer_sJXd · 2024-11-03

**Soundness:** 3
**Presentation:** 2
**Contribution:** 3
**Rating:** 8
**Confidence:** 4

**Summary:**

This paper introduces Provence, a context pruner for question answering that formulates context pruning as a sequence labeling task. Specifically, the authors fine-tune a pre-trained re-ranker (DeBERTa) using silver labels generated by Llama-3-8B. The claimed benefits of this approach are that the proposed approach operates without assuming a fixed compression ratio and can be applied out-of-the-box to various datasets. This approach improves the efficiency of RAG for QA without performance degradation.

**Strengths:**

- Provence is a novel approach that enables context compression without performance drop, outperforming baseline models.
- It offers practical advantages, including query-dependent, sentence-level, dynamic compression, and adaptability across multiple domains.
- The authors conduct extensive analysis to understand the properties of Provence and explore various design options to enhance its effectiveness.

**Weaknesses:**

- Some hypotheses are presented without supporting results. For example: (1) Provence does not select any irrelevant sentences, and (2) leftmost and rightmost sentences are usually not deemed relevant and no explanation is provided for why catching “needle” sentences in these positions might be an exception.
- There is insufficient information to indicate that the experiments were designed for a fair comparison. The authors do not consider an adequate retrieval pipeline for comparison with existing approaches (as Splade + DeBERTa may introduce bias favoring Provence), and the impact of the hyperparameter k in retrieving the top-k relevant passages is not discussed.

**Questions:**

- Why is it useful to unify compression and reranking? In which parts of a RAG pipeline can rerank scores be effectively used? Does Provence outperform existing approaches in reranking?


-----
Raised my score 6-> 8 as author responses cleared most of my doubts.

---

> ### Author Response · Authors · 2024-11-20
> **Author response**
>
> Dear Reviewer, many thanks for your insightful feedback! We would also like to thank you for __highlighting the important advantages of our method, such as being query-dependent, sentence-level, having dynamic compression, and adaptable across multiple domains__!
>
> We would like to clarify some of your concerns below.
>
> __Weakness 1.1: “Provence does not select any irrelevant sentences”__
>
> This phrase __has a prefix “In most cases”__. In the initial version of pdf, this phrase was in line 444 and the prefix was in line 423; where lines 424-443 in between were footnotes and the figure caption; in the newest version this sentence is positioned better. Furthermore, this line is a part of a description of a __toy experiment__ with synthetically built contexts. We do not claim that Provence never selects any irrelevant sentences at all.
>
> In this toy experiment, reported in Figure 3 (left), for the middle range of positions in text (meant by “in most cases”), Match is close to 100% and the number of selected sentences equals to the ground truth number of sentences. This can only be the case if no irrelevant sentences are selected (in most cases), by the design of the toy experiment.
>
> __Weakness 1.2: “leftmost and rightmost sentences are usually not deemed relevant and no explanation is provided”__
>
> Thank you for noting this, we agree that this could indeed be extended further.
>
> First, we __updated the corresponding Figure 3 (left) in the pdf with the better results__, after fixing a small issue we found in our code. The issue was that we were removing from the training data any example with the single left-most relevant sentence; it should not have affected Provence performance beyond this case. In the updated version, the 1-sentence left-most needle now gets selected much more often than in the original version. We also used a newer model with additional data augmentation (the performance of the model on other evaluations is identical).
>
> Second, let us expand the discussion of the leftmost and rightmost needles, where indeed we observe some _occasional_ errors of context pruning. The reason for the drops is that training data has little examples of the corresponding types of relevant sentences. For example, it can be seen for the rightmost position in the statistics Figure 5, right in the Appendix. We plan to work on further improving these positions in future works.
>
> We would like to note that the drop in the left most position is not a problem because in all other evaluations we always select the first title sentence, as described in lines 346-348.
>
> __Weakness 2: experiments design and fairness__
>
> We did our best to __thoroughly design our experiments and provide support to our claims__. We rely on the setups recommended in [1] as a strong RAG baseline, relying on the strong retrieval pipeline, including reranking __(often ignored in the literature)__, and using LLM-as-a-Judge (aka LLMeval) metric which is recommended by [1] and, according to [4] (Fig 3b), __correlates better with human judgements__, compared to widely used match-based metrics.
>
> > an adequate retrieval pipeline for comparison with existing approaches (as Splade + DeBERTa may introduce bias favoring Provence)
>
> In Figure 2, we include the results with the Deberta reranker, to ensure that all approaches are reranked in the same way. Since Provence (with reranker) is relying on Deberta, if we used other rerankers for other approaches, it could cause additional differences in performance.
>
> At the same time, __in Appendix Figure 8, we test Provence in other settings, including other retrieval pipelines and other generators__. To better address your concern about bias towards Deberta, we also added the results for RetroMAE retrieval + BGE reranking (both models changed compared to the setting of Figure 2), demonstrating __the same Provence behavior as in the main plot__.  Furthermore, we added Figure 9 which compares Provence to different context compression baselines in the RetroMAE retrieval + BGE reranking setting. This Figure is very similar to Figure 2 and suggests that  __Provence robustly outperforms other baselines regardless of retrieval/reranker settings__.
>
> __We hope these updates to the results resolve your concerns about our experimental settings!__
>
> > the impact of the hyperparameter k in retrieving the top-k relevant passages is not discussed
>
> We added Figure 10 in Appendix, testing Provence with top-k = 1, 5, and 20. In all cases the performance of Provence is identical to the behavior shown in the main plot in Figure 2.

---

> ### Author Response · Authors · 2024-11-20
> **Answer to the questions**
>
> Below we would like to provide answers to your Questions.
>
> > Q1 Why is it useful to unify compression and reranking? In which parts of a RAG pipeline can rerank scores be effectively used?
>
> As described in lines 125-127 of Introduction, reranking is a second stage of retrieval. The first stage retrieval encodes queries and passages independently for efficiency, i.e. passages are pre-encoded offline, and when a user asks a question, only the question gets encoded and the fast search is being performed. The second stage, reranking, operates on top of the results of the first stage (hence a much smaller number of passages, e.g. 50), and encodes each context _together_ with the query. This provides much more informative embeddings and substantially improves search results, [2,3]. __To summarize, reranking is an essential part of modern search.__
>
> Recently,  [1] showed that reranking (often overlooked by the NLP community) does systematically improve the performance of RAG pipeline and leads to a strong baseline.
>
> __Our novel contribution__ is to __complement reranking, an already present part of the search in RAG,  with context pruning capabilities__, because these models have the same input.
> Thus, Instead of having a pipeline query>>retrieve>>rerank>>prune>>generate, we can do query>>retrieve>>provence>>generate, which means that __pruning comes at no extra computational cost__.
>
> > Q2 Does Provence outperform existing approaches in reranking?
>
> As shown in Table 4 in the main text, __Provence performs the same in reranking__ (in domain and out of domain) as the initial reranking model we start from. We do not have the goal of producing a better reranker; instead we __incorporate new capabilities (context pruning) into reranking__, an existing stage in the RAG pipeline, __without losing its performance__.
>
> [1] BERGEN: A Benchmarking Library for Retrieval-Augmented Generation. Rau et al. EMNLP 2024 https://arxiv.org/abs/2407.01102
>
> [2] Learning to Rank for Information Retrieval, Liu Book 2011
>
> [3] Pretrained Transformers for Ranking: BERT and Beyond Lin et al. 2020 https://arxiv.org/abs/2010.06467
>
> [4] Judging the Judges: Evaluating Alignment and Vulnerabilities in LLMs-as-Judges. Thakur et al. July 2024

---

> > ### Author Response · Authors · 2024-11-25
> > **Discussion period end approaching**
> >
> > Dear Reviewer, we thank you again for the valuable feedback which allowed us to improve our work. We hope you had time to check our __updated pdf, comments and clarifications__.
> >
> > We sincerely believe that __we have been able to address your concerns listed in the review__: (1) improving the results and the discussion of the needle-in-the-haystack experiment, (2) providing evaluation results in settings with a different retriever-reranker pair and with various top-k, (3) clarifying the contribution about the integration of context pruning and reranking.
> >
> > Since we are approaching __the end of the discussion period__, we would appreciate it if you could __provide us feedback__ on our response and reflect it in __the assigned scores and the overall assessment__. We are also happy to answer any further questions.

---

> > > ### Comment · Area_Chair_aQEp · 2024-11-26
> > >
> > > Dear Reviewer sJXd, the ICLR discussion period is extended. Could you please take a look at the authors' rebuttal and other reviews, and see whether you would like to update your ratings? The authors would greatly appreciate your consideration and responses.

---

> ### Comment · Reviewer_sJXd · 2024-11-28
> **Response acknowledgement**
>
> Dear authors, Thank you for your effort in responding, and I'm sorry for the late acknowledgment. I've taken some time to go through the paper again to refresh my memories.
>
> - I think the question regarding the dependency on certain reranker (DeBERTa) models/architecture has been partially resolved. One more reranker (BGE) has been tested, although I am not entirely sure why the retriever was changed. Maybe you can test it on different combinations of retriever & reranker now as they are already trained, it should be just plug-and-play.
> - I am not too sure about weakness 1.2. I think position dependency is a weakness. I am unsure why authors take the whole set of passages together as an input rather than processing all sentences independently (in parallel).
>
> While rereading your paper again, I understood the essence of PROVENCE is at "reranking + pruning" at the same time (thus more efficient). I have two extra categories of questions **(1) Comparison to the recent RAG paper** and **(2) Evaluation with LLMs**. I understand these questions can be long, but since we are given a bit more time, I think answering these questions can be helpful in precisely positioning your paper alongside existing work.
>
> ### (1) Comparison to recent RAG paper: RE-RAG[1].
>
> Your paper reminded me of the paper at EMNLP2024 "RE-RAG"[1] as its main claim was also that their relevance estimator (RE) module can rerank and classify whether the document is relevant to the query or not at a sentence level.  [1] does not use RE in pruning, but shows classification accuracy as well as different decoding strategies when all documents are deemed to be irrelevant.
>
> - As PROVENCE largely selects a whole chunk of sentences (following 268-297), I feel this paper is somewhat similar or that the RE module in RE-RAG can do a similar job.  (although PROVENCE has potential to select at token level it doesn't seem to do so right now)
> - I felt the fact RE module uses smaller models such as T5-base, large (smaller than LLMs as PROVENCE does), and plug-and-plays to frozen LLMs, I wanted to see what maybe the core difference between the two besides the fact that they are used in a different fashion.
> - PROVENCE does not improve reranking but incorporates only pruning into existing reranked if I understood correctly. RE and many other rankers do that. What is your opinion on this? If the reranker + sentence selection can improve the original reranker and select sentence, would you say this is different?
> >" Provence performs the same in reranking (in domain and out of domain) as the initial reranking model we start from. We do not have the goal of producing a better reranker; instead we incorporate new capabilities (context pruning) into reranking, an existing stage in the RAG pipeline, without losing its performance."
> - I understand RE-RAG isn't exactly in the same setting; however, would there be any way to compare this model? It would be nice to have it on an experimental level, but if not, perhaps on a conceptual level. I think the motivations for the training steps are also similar in some sense. PROVENCE took powerful LLM's capability, and RE-RAG took T5's gold answer probability given contexts.
>
> [1] Kim & Lee., RE-RAG: Improving Open-Domain QA Performance and Interpretability with Relevance Estimator in Retrieval-Augmented Generation, EMNLP 2024
>
> ### (2) Evaluation with LLMs.
>
> - I see most of the accuracy evaluations are done through LLMeval. For many ODQA datasets, often EM and accuracy (whether the answer is included) are reported. For a fair comparison with existing literature in ODQA & RAG, is it possible to provide these statistics as well?
>  - I believe this won't be too much effort as the prediction results are already in the log.
> - I am asking this question as colleagues of mine and I tried utilizing LLMeval in order to collect silver data in similar sense you did, and we observed that it was actually too lenient as it would often mark a predicted answer that has few words overlap with gold as correct.
>
> Sorry for putting a lot of questions near the end. When I was writing the initial review, I did not recognize the resemblance with the paper I recommended comparing. I am keeping my score for now as I agree with the aim and direction that the paper presents.

---

> > ### Author Response · Authors · 2024-12-02
> >
> > Dear reviewer,
> >
> > Thank you for engaging with our rebuttal!
> >
> > We would like to note that the reviewer’s comment was posted on November 28, during the last period of the author-reviewer discussion. Since this period of discussion is devoted to clarification questions, and given that __we are not allowed to update the pdf anymore__, __we will not be able to provide new results__.
> >
> > >  test it on different combinations of retriever & reranker
> >
> > We will add the results with more combinations of retriever & reranker in Figure 8 in Appendix, including the SPLADE & BGE-M3 combination. We also checked the performance with 1) an additional reranker ( “cross-encoder/ms-marco-MiniLM-L-6-v2”) 2) SPLADE & BGE-M3, and they are identical to all the six other settings reported in Figure 8.
> >
> > We would like to highlight that the current version of the paper already demonstrates the high performance of Provence with 2 different retrieval-reranking settings. Moreover, on the RGB dataset, we use oracle documents, and in Figure 3 (middle) we demonstrate that in case of randomly selected contexts (completely irrelevant to the query), Provence almost always selects 0 sentences (empty context). All these results together demonstrate Provence robustness to the used retrieval setting.
> >
> > > I am not entirely sure why the retriever was changed
> >
> > Our motivation to change both models was to demonstrate the high performance of Provence in a setting with all models being different from the training setting.
> >
> > > I am unsure why authors take the whole set of passages together as an input rather than processing all sentences independently (in parallel).
> >
> > In many cases, sentences have __references to previous sentences__, i.e. __previous sentences are necessary to understand the current sentence__. See, for example, footnote 7 in the main text:
> >
> >
> > ```
> > Question: “Which library was used in the experiments?””
> > Context: “Experiments were conducted using a library. __Its__ name is Bergen.”
> > ```
> >
> > This example __will not be answered correctly if we process sentences independently__, because “Its name is Bergen” does not contain information that Bergen is a library.
> >
> > Another example from the NQ dataset is presented in Appendix Table 7:
> >
> > ```
> > “Shepherd’s pie. History. In early cookery books, __the dish__ was a means of using leftover roasted meat of any kind, and the pie dish was lined on the sides and bottom with mashed potato, as well as having a mashed potato crust on top. <...> In Quebec, a variation on the cottage pie is called ”Patˆe chinois”.  __It is__ made with ground beef on the bottom layer, canned corn in the middle, and mashed potato on top..”.
> > ```
> >
> > If we process sentences independently, it is unclear what “the dish” and “it is” refer to.
> >
> > These are the simplest examples. In practice, coreferences can be much more complicated, e.g. example from Table 8 in Appendix:
> >
> > ```
> > ”A number of plant species produce glycosides that are sweet at concentrations much lower than sugar. The most well-known example is glycyrrhizin, the sweet component of licorice root, which is about 30 times sweeter than sucrose. __Another commercially important example__ is stevioside, from the South American shrub ”Stevia rebaudiana”.
> > ```
> >
> > Without previous sentences, it is unclear an example of what is “stevioside”.
> >
> > To summarize, __coreferences between sentences happen all the time in natural language__ and may have various forms. Provence encodes all sentences together, to provide __rich semantic representations__ and enables __high quality semantic pruning__, __substantially outperforming approaches which encode sentences independently__, e.g. RECOMP and DSLR (Figure 2).
> >
> > > position dependency is a weakness
> >
> > We would like to highlight that __in practice this weakness is almost not pronounced__, because (1) the rightmost position happens to be relevant very rarely (shown in Figure 5, right in Appendix); (2) the leftmost position is always selected by default in our implementation (lines 346-348); (3) this weakness is only visible in the specifically designed experiment in Figure 1 and not in the main Figure 2 (performance with Provence pruning identical to the performance with full context).
> >
> > We also keep working on improving the processing of these positions, e.g., we already improved their processing in the most recent version of the pdf.

---

> > > ### Author Response · Authors · 2024-12-02
> > >
> > > ### (1) Comparison to the recent RAG paper
> > >
> > >
> > > Thank you for pointing out the Re-RAG paper! We would like to highlight that __Re-RAG and Provence address different problems__.
> > >
> > > Re-RAG proposes an alternative approach to reranking and also a way of __utilizing reranking scores__ in generation to __improve generation performance__. At the same time, Re-RAG __does not modify retrieved passages and does not address RAG efficiency__.
> > >
> > > On the contrary, Provence relies on __existing reranking approaches__ and proposes to __reduce the size of the retrieved passages__ (context pruning) which leads to __speed up in generation__ (improving efficiency).
> > >
> > > To summarize, Provence and Re-RAG address different problems, and these approaches can be complimentary, e.g., reducing the size of retrieved contexts with Provence for faster generation and utilizing reranking scores from Re-RAG in generation for higher-quality predictions. Provence could also be in principle trained from the pretrained RE, although __we were not able to find a publicly released RE model__.
> > >
> > > >  PROVENCE largely selects a whole chunk of sentences
> > >
> > > Provence selects __entire__ sentences, but __not all sentences__ in the context, e.g., it usually selects a subset of sentences.
> > >
> > > > PROVENCE does not improve reranking but incorporates only pruning into existing reranked if I understood correctly. RE and many other rankers do that. What is your opinion on this?
> > >
> > > As explained above, the RE module does not perform context pruning and does not modify retrieved contexts at all, and neither do any other existing rerankers, to the best of our knowledge.
> > >
> > >
> > > ### (2) Evaluation with LLM
> > >
> > > > For a fair comparison with existing literature in ODQA & RAG, is it possible to provide these statistics as well?
> > >
> > > We would like to note that Figure 7 in Appendix already presents the results with Recall, a match-based metric which measures the percentage of tokens from the short gold answer that are present in the generated response. We chose this match-based metric over Match or EM because they are too restrictive and lead to a lot of false negatives.
> > >
> > > To better address your concern and make our results comparable to other papers, __we will release a zip archive__ with generations from all the models in Figure 2 and __json files listing all metrics: Match, recall, and LLMEval__. This zip archive will be a part of the code release for the paper.
> > >
> > > For example, below we compare numbers from the Re-RAG paper, our baseline from Bergen (full context), and Provence (after context pruning with T=0.1). Here metric is Accuracy (whether the answer is included):
> > >
> > > |                            | NQ (accuracy)         | NQ (accuracy)         |
> > > |----------------------------|-----------------------|-----------------------|
> > > |                            | Generator: LLama-2-7b | Generator: LLama-3-8b |
> > > | RE-RAG                     | 48.4                  | 54.4                  |
> > > | Bergen (full context)      | 64.6                  | 67.6                  |
> > > | Provence (pruned contexts) | 64.8                  | 67.2                  |
> > >
> > > > I am asking this question as colleagues of mine and I tried utilizing LLMeval in order to collect silver data in similar sense you did, and we observed that it was actually too lenient as it would often mark a predicted answer that has few words overlap with gold as correct.
> > >
> > > We do not observe this effect since in our implementation we do not use any gold answers in silver data generation, i.e., an LLM is asked to highlight relevant parts only given a question and context.
> > >
> > > > since we are given more than a week, I think answering these questions can be helpful in precisely positioning your paper alongside existing work
> > >
> > > This timing is not exactly correct :) The reviewer’s comment was posted on November 28. leaving us only 4 days (until Dec 3) to reply (including 2 weekend days, or only 2 working days), or even 3 since we want to leave some time for the reviewer to answer us.
> > >
> > >
> > > Thank you again for engaging with our rebuttal and hope that our response resolves your remaining concerns. In that case, we would appreciate it if you consider increasing your scores.

---

> > > > ### Comment · Reviewer_sJXd · 2024-12-02
> > > > **Raising my score as most of my doubts were cleared away**
> > > >
> > > > > This timing is not exactly correct :) The reviewer’s comment was posted on November 28. leaving us only 4 days (until Dec 3) to reply (including 2 weekend days, or only 2 working days), or even 3 since we want to leave some time for the reviewer to answer us.
> > > >
> > > > Thanks for the clarification. Sorry for posing hard questions last minute, but I think it was worth asking reading your answers makes me understand the settings better. I hope the subsequent readers and other reviewers &  AC can benefit form this as well.
> > > >
> > > > > We do not observe this effect since in our implementation we do not use any gold answers in silver data generation, i.e., an LLM is asked to highlight relevant parts only given a question and context.
> > > >
> > > > I see, however, I think it might be nice to have some evaluation on this matter. I understand that your final metric is most important, but as other reviewer has asked as well,  some analysis might be helpful.
> > > >
> > > > > To better address your concern and make our results comparable to other papers, we will release a zip archive with generations from all the models in Figure 2 and json files listing all metrics: Match, recall, and LLMEval. This zip archive will be a part of the code release for the paper.
> > > >
> > > > I think some way of displaying the accuracy results (like your Bergen v.s. Provence results) would be useful. Maybe just as a table in appendix? Of course, your suggestion of zipping the log sounds really nice as well.
> > > >
> > > > ### Questions about RE-RAG
> > > >
> > > > Your point about not being able to compare with RE-RAG as there is no code released makes sense and is understandable. However, the table presented above doesn't mean much I think as they are utilizing completely different baseline retriever.
> > > >
> > > > ### Your clarification about RE module and Provence
> > > >
> > > > > Provence selects entire sentences, but not all sentences in the context, e.g., it usually selects a subset of sentences.
> > > >
> > > > I think this clarified most of my doubts. I misunderstood the sentences in L.268-297, thinking that the whole paragraph was selected. So, the clarification that the retrieved chunk consists of sentences was helpful. Although it was straightforward when first reading the paper, I admit that I got confused in the middle.
> > > >
> > > > I think author's responses mostly make sense and I am happy to raise my score.

---

> > > > > ### Author Response · Authors · 2024-12-03
> > > > > **Thank you!**
> > > > >
> > > > > Thank you for your feedback and the score increase!
> > > > >
> > > > > We appreciate your effort and time reviewing our work! We will certainly take your comments into account in the next revision.

---

### Official Review · Reviewer_99vA · 2024-11-03

**Soundness:** 3
**Presentation:** 3
**Contribution:** 3
**Rating:** 6
**Confidence:** 4

**Summary:**

The paper proposed a context pruning method for retrieval augmented generation tasks by considering the context pruning task as passage reranking. The authors used a Deberta reranking model which is significantly smaller than large generative models to improve both efficiency and performance of retrieval augmented language generation.

Experiments show that when the compress rate is higher than 50%, the proposed method outperforms other strong baselines. The authors also provided comprehensive analysis and ablation studies.

**Strengths:**

The proposed method shows a good adaptation ability and efficiency on different tasks. When the compression rate is higher than 50%, the method still achieves a reasonable performance.

**Weaknesses:**

This work misses some relevant citations and does not compare to these methods, for example:

[1] DSLR: Document Refinement with Sentence-Level Re-ranking and Reconstruction to Enhance Retrieval-Augmented Generation.

[2] Dense X Retrieval: What Retrieval Granularity Should We Use?

[3] Phrase Retrieval for Open-Domain Conversational Question Answering with Conversational Dependency Modeling via Contrastive Learning

[4] Decontextualization: Making Sentences Stand-Alone

I'll adjust my score if the authors settles my concerns about missing these comparisons.

**Questions:**

what is the performance of the proposed method when the compress rate is less than 50%?

---

> ### Author Response · Authors · 2024-11-20
> **Author response**
>
> Dear Reviewer, many thanks for your insightful feedback!
>
> ## Addressing Weaknesses and Questions
>
> Thanks for pointing us to these related works! We address those in the following:
>
> __[1] DSLR: Document Refinement with Sentence-Level Re-ranking and Reconstruction to Enhance Retrieval-Augmented Generation.__
>
> Please note that this paper is released on arXiv on July 4 which is later than ICLR __concurrent work__ cut-off of July 1 (https://iclr.cc/Conferences/2025/FAQ). Nonetheless, we did run an experiment with DSLR, with our DeBERTa-based reranker and added it in Figure 2. The results show __DSLR is indeed a strong baseline__, however we find  that __Provence is more effective than this baseline.__ __Provence prunes sentences considering their context__ and is __trained for the  context pruning task__, in contrast to __DSLR which processes sentences independently__ and uses __off-the-shelf rerankers__ which may be not adapted to sentence-level processing.
>
> __[2] Dense X Retrieval: What Retrieval Granularity Should We Use?__
>
> This paper proposes to reformulate passages into propositions and run retrieval on the propositions level. We would like to note that this approach requires __a very time consuming preprocessing__, which may be even infeasible in practice for large datastores. At the same time, our approach, __Provence with reranking, has an almost zero cost__, as context pruning is integrated into the reranking stage of a standard RAG pipeline.
>
> Since the authors release the preprocessed Wikipedia datastore, we added the results with it in Figure 2. The compression rate is computed by comparing the length of top-5 retrieved propositions to the length of top-5 retrieved passages. We find that __this approach substantially underperforms compared to Provence__. In manual inspection of a subset of examples, we found that sometimes propositions lose some details which were present in the passages and which are essential to answer given questions.
>
> __[3] Phrase Retrieval for Open-Domain Conversational Question Answering with Conversational Dependency Modeling via Contrastive Learning.__
>
> This paper is about Phrase Retrieval for Conversational QA. We believe that is hard to compare this paper to our setups: a) the method is proposed for conversational QA, which is __a different task from single-turn QA that we focus on__; b) the models used rely on finetuning of the LLMs (reader), which is __different from the setting we consider, i.e. zero-shot RAG__; c) short answers are expected from the model, which is also a different setting compared to ours.
>
> __[4] Decontextualization: Making Sentences Stand-Alone__
>
> This paper argues that sentences can be reformulated to better reflect their context when picked in isolation. In a RAG setting, this leads to something somewhat similar to Dense X retrieval, where we could retrieve from a collection of contextualized sentences (instead of passages). The __scarcity of aftefacts__ released by the paper makes it hard to compare to their work, however we believe that __comparison to a more recent Dense X retrieval is sufficient to represent this group of methods__ in our experiments.
>
> > what is the performance of the proposed method when the compress rate is less than 50%?
>
> The task performance gets close to the full context performance; we added such points in App. Figure 8 (gray).
>
> ## Clarification of summary and strengths
>
> We would like to bring a clarification into the reviewer’s summary, which hopefully could be taken into account to reconsider the contribution assessment of our work.
>
> > “The paper proposed a context pruning method for retrieval augmented generation tasks by considering the context pruning task as passage reranking”
>
> Our approach does not cast context pruning as reranking but __unifies context pruning and passage reranking in a single model__, enabling __almost zero cost context pruning__. Context pruning is performed using a specifically developed approach based on sequence labeling, which enables __adaptability__ to various amounts of relevant sentences in a passage and __robustness__ to different domains.

---

> > ### Comment · Reviewer_99vA · 2024-11-24
> > **Helpful clarifications**
> >
> > Thank you for providing the comparisons and the updated manuscript! I think the clarifications are helpful and the updated experiment results are convincing. As a result I'll update my scores.

---

> > > ### Author Response · Authors · 2024-11-25
> > >
> > > Dear Reviewer,
> > >
> > > Thank you a lot for taking a look at our response and updating your scores!
> > >
> > > Considering that the review does not appear to raise additional concerns about the paper’s acceptance, we wonder if it might be possible to reconsider the overall assessment from "marginally above threshold" to a higher score.
> > >
> > > Thank you again for your consideration.

---

### Official Review · Reviewer_7vVg · 2024-11-04

**Soundness:** 2
**Presentation:** 1
**Contribution:** 2
**Rating:** 3
**Confidence:** 3

**Summary:**

The work presents a context pruning technique, where during RAG, each sentence in the context is individually labeled as relevant or irrelevant to the query. The training is conducted by distilling from Llama-3-8B, where the model is instructed to output an extractive summary on the sentence-level of the context given some query. Note that though the labeling is on the sentence level, each token in the sentence will inherit the sentence's label during training. During inference, only sentences where +50% tokens are kept are considered kept/positive. In another variant, the pruning model is used to both output a binary judge mask and serve as a reranker by outputting a relevance score, though for this case the model needs to be initialized from pretrained reranker. The proposed model achieves competitive results on diverse QA tasks, most on par with strong baselines like RECOMP but with more compression ratios.

**Strengths:**

• Evaluated on diverse QA tasks, making the results/improvements less likely to be cherry-picked

**Weaknesses:**

• It is questionable for a non-instruction-tuned model like DeBERTa to understand slightly more complex relation definitions. For example, if the query asks DeBERTa to find sentences that support some point vs. refute the same point, chances are that DeBERTa is going to keep the same sentences.

• Results figure (i.e., Figure 2) is not clear. Not clear what x- & y-axis represent. The caption should clearly state that, e.g., y-axis is compression ratio, etc...

• Overall results figures are all hard to read, esp. Figure 4 where all symbols are superimposed on top of each other. Consider enlarging them or changing them to more informative tables.

• I don't quite get the model choices here. The silver labelings are obtained with Llama-3 models, where the generator is chosen to be Llama-2? Is there any justification for the versioning discrepancy?

**Questions:**

Suggestions:

• 181: using "silver labels" without defining the term first. From the context it seems to mean "auto-generated labeled data".

---

> ### Author Response · Authors · 2024-11-20
> **Author response**
>
> Dear Reviewer, many thanks for your insightful feedback! We would like to clarify some of your concerns below.
>
> ## Addressing Weaknesses
>
> First of all, thanks for pointing out which presentation parts were not clear, we corrected them in the pdf:
> * We updated the captions and axes labels in the figures [addressing Weakness 2]
> * We added Tables in Appendix, duplicating scores reported in Figures in the numeric form [addressing Weakness 3]
> * We clarified the definition of silver labels [addressing Suggestion]
>
> __Weakness1:  non-instruction-tuned model vs  complex understanding__
>
> Thank you for suggesting a very interesting example.  __Provence is not designed and not expected to operate on such complex semantic understanding__. In the given example, Provence will keep both sentences since they both are _related to the query_. Provence is designed  to filter out sentences which are _unrelated to the user query_, which happens very often due to passage-level retrieval and enables substantial compression and generation speed up, as shown in Figure 2 and Table 3. This is __a simpler task for which DeBERTa is well suitable__, while more complex reasoning will be performed on the generation step by an LLM.
>
> Example:
>
> Question: _Can you eat pumpkin every day?_
>
> Context with [ ] highlighting a fragment selected by Provence:
>
> _Pumpkin is at its peak in the fall. [Eating pumpkin every day can help keep you regular, reduce inflammation, strengthen your immune system and promote eye health. It may also help lower blood pressure. However, if you eat too much, you may experience diarrhea from a high dose of fiber.] Read on to learn all about pumpkin’s nutrition and health benefits._
>
> Provence selected arguments for and against eating pumpkin every day and removed other sentences. Here we provide a minimal example, in practice the amount of removed sentences is higher on average, as reported in Figure 2.
>
> This example also demonstrates well __a distinguished design property of Provence__, namely __automatically detecting the number of relevant sentences in the context__, based on a chosen threshold. In this example, Provence correctly detects that 3 sentences need to be selected. E.g. RECOMP (ext) requires specifying the number of relevant sentences as a hyperparameter, which is very limiting in practice (as it depends on each passage etc.)
>
> __Weakness 4: Model Choice (LLama-2)__
>
> Provence is supposed to work well out-of the box to be used with any LLMs, so there is __no requirement on the oracle labeler and the generator being the same LLM__.
>
> Our motivation to run main experiments with a __weaker generator__ is to create a more __challenging setting for context pruning__. A strong generator relies more often on its internal knowledge rather than a provided context, hence mistakes in context pruning would be less pronounced. We included this motivation in the updated pdf (footnote 3 on page 7).  We choose LLama-2-7b-chat as this weaker generator, due to its common use in the literature.
>
> At the same time, as referenced in line 340 (333 in prior version) in the main text, __we include results with stronger generators in Appendix Figure 8__. Inspired by reviewers suggestions, we now further extended this Figure, by including even more settings and all datasets. For example, this Figure demonstrates the same behavior of Provence, as shown in the main text, for such generators as Solar-10.7B, Mistral-7B-instruct, or LLama-3-8B. Results hold for all the generators, highlighting the robustness of Provence.
>
> ## Clarification of summary and strengths
>
> We believe there might also be some misunderstanding of our contribution, based on the reviewer’s summary, which we would like to address below.
>
> > The work presents a context pruning technique, where during RAG, each sentence in the context is __individually__ labeled as relevant or irrelevant to the query
>
> The word “individually” seems misleading in this sentence, since __the main essence of Provence is encoding all sentences in the context together__, so that the representations of sentences are contextualized on each other and hence labeling each sentence as relevant or irrelevant also depends on other sentences.
>
> > The proposed model achieves __competitive results__ on diverse QA tasks, most on par with strong baselines like RECOMP but with more compression ratios.
>
> Provence __outperforms__ all other methods, by achieving substantially higher task performance on most of the datasets than other approaches, including RECOMP, with the same compression ratio, as reported in Figure 2 (Provence consistently being on the Pareto front). This is noted e.g. by Reviewer sJXd.
>
> Finally, we believe that __the review does not fully reflect the advantages of our method__, outlined in lines 99-106 in Introduction, namely developing a novel context pruning method which is __adaptable, robust and efficient__.

---

> > ### Author Response · Authors · 2024-11-25
> > **Discussion period end approaching**
> >
> > Dear Reviewer, we thank you again for the valuable feedback which allowed us to improve our work. We hope you had time to check our __updated pdf, comments and clarifications__.
> >
> > We sincerely believe that __we have been able to address your concerns listed in the review__: (1) improving paper presentation, (2) providing clarifications for the considered task and for the choice of the LLMs. We also provided __clarifications to the contributions of our work__, since we believe they are not fully reflected in the current version of the review.
> >
> > Since we are approaching __the end of the discussion period__, we would appreciate it if you could __provide us feedback__ on our response and reflect it in __the assigned scores and the overall assessment__ in light of this rebuttal and the other reviews. We are also happy to answer any further questions.

---

> > > ### Comment · Area_Chair_aQEp · 2024-11-26
> > >
> > > Dear Reviewer 7vVg, the ICLR discussion period is extended. Could you please take a look at the authors' rebuttal and other reviews, and see whether you would like to update your ratings? The authors would greatly appreciate your consideration and responses.

---

### Official Review · Reviewer_5r2e · 2024-11-09

**Soundness:** 3
**Presentation:** 4
**Contribution:** 3
**Rating:** 8
**Confidence:** 4

**Summary:**

This paper presents an encoder-based context pruning approach to enhance retrieval-augmented generation.
The proposed framework aims to identify and eliminate irrelevant portions of retrieved context to improve the accuracy of generated responses.
Unlike previous methods that typically rely on LLM-like decoder models, the proposed approach leverages an encoder model to frame the problem as a sequence labeling task, similar to POS tagging and NER.
The authors generate training data (silver-standard datasets) using another LLM, such as LLaMA-3.2-8B, and evaluate the performance of context pruning techniques with LLMs serving as evaluators.
The experimental results demonstrate that the proposed method is both effective and efficient.
A comprehensive set of analyses further validates the impact of the approach.

**Strengths:**

- The draft is well-structured, making it easy for readers to grasp the core concepts. I genuinely enjoyed reading it.
- Unlike previous approaches that employ LLMs, the proposed method is specifically optimized for the target task, allowing it to achieve both effectiveness and efficiency.
- A series of diverse experiments highlight the merits of the proposed approach from various angles, instilling confidence in readers about the method’s reliability.

**Weaknesses:**

- Training and evaluation procedures rely heavily on the use of LLMs. This is a bit concerning for me, for the following reasons:
    - Firstly, if we already have a well-performing LLM capable of effective context pruning, why should we develop a separate model for this task? The strong performance of the existing LLM could suggest that it inherently possesses the ability to filter and focus on crucial information from the context. Rather than adding an external module, we might leverage advanced prompting techniques to enhance this functionality.
    - Secondly, the process of training and evaluation lacks human oversight. Can we be confident that a model trained and evaluated in this way will align well with human preferences? Although LLMs are initially trained on human-generated data, I wonder if there’s a risk of a “hacking” effect, where the model is tuned to satisfy only the evaluation metrics instead of genuinely aiming to achieve the intended goal.
- Figure 2’s main results are intuitive; however, this presentation style may open the door to different, subjective interpretations among readers. For instance, if a reader prioritizes compression rate, they may see RECOMP (abs) as the most appealing option. Thus, it may be helpful to introduce a unified metric that incorporates both compression ratio and LLM evaluation performance, potentially using a measure akin to AUROC.

**Questions:**

- Given the importance of both performance and compression rate, exploring a dynamic trade-off between the two could be beneficial. For instance, while one user may prioritize performance over compression, another might value compression more highly. A straightforward way to enable this dynamic trade-off could involve adjusting the threshold that determines the relevance of tokens. By fine-tuning this threshold, the model could flexibly adapt to user-specific preferences, optimizing either performance or compression as needed.

---

> ### Author Response · Authors · 2024-11-20
> **Author response**
>
> Dear Reviewer, many thanks for your good feedback. We would also like to thank you for __highlighting the important advantages of our work, such as developing a method  which is specifically optimized for the context pruning task, allowing it to achieve both effectiveness and efficiency__!
>
> Below are some answers to your comments and questions.
>
> > Weakness 1.1 (What if we have a very good LLM): Efficiency
>
> 1. Even though modern LLM can reason over quite a long context, and implicitly filters out irrelevant information, longer context processing induces __computational overhead__. Our results in Table 3 demonstrate that by compressing __with Provence we can have 2x speed up at inference.__
> 2. If we wanted to use such strong LLM for context compression this would be less efficient compared to our small  model: Table 2  reports context processing time required by different models. Usage of LLama3-8b would be somehow equivalent to LongLLMLingua in processing time and MFLOPS required, which is __far more expensive than our method__.
>
> > Weakness 1.2 : training and evaluation lacks human oversight, “hacking” effect
>
> In training: (1) The oracle labeling task (select context relevant for the query) is quite different from the evaluation task (judge whether an LLM-generated answer is similar to the reference answer), therefore it is hard to see how training one  could optimize the other (“hacking effect”).
> (2) Furthermore, we decouple the oracle labeler model (Llama3-8b) from the generator (LLama2-7B). Extra runs with other diverse models (SOLAR, Mistral, etc., Appendix Figure 8) confirm that the context compressor is robust across various generators.
>
> As for the evaluation, it is indeed very important topic, and proper evaluation of RAG systems is an open question. While LLM-based evaluation is not perfect and may be prone to occasional errors, current studies [1, 2] suggest that LLMEval is still more robust compared to lexical metrics which are often used for RAG evaluation.  We also report the results with match-based metrics in Appendix Fig. 7.
>
> >W2 Unified Metric:
>
> Thanks for your remark, we agree that it could be interesting to develop such a metric. However, one needs to weigh the importance of compression vs accuracy accordingly, which is not straightforward.  In [3] such a metric is proposed for information retrieval by allowing users to weight hardware cost, latency and accuracy of different models to establish the final leaderboard. That would be an interesting direction to explore in the future work.
>
> In a meantime, we would like to argue that presenting the results as a __Pareto front is more informative__  and more scientifically sound, allowing the interested researcher to make an informed decision within ones' efficiency/effectiveness preferences.
>
> > Question
>
> We are not sure we understood your question, but indeed, as noted in footnote 2, a user can further adjust a threshold to their needs, prioritizing compression or performance. If you mean a possibility of automatically adjusting the threshold, we agree it would be an interesting direction for future research.
>
> [1] Evaluating Open-Domain Question Answering in the Era of Large Language Models. Kamaloo et al., ACL 2023
>
> [2] Judging the Judges: Evaluating Alignment and Vulnerabilities in LLMs-as-Judges. Thakur et al. July 2024
>
> [3] Moving Beyond Downstream Task Accuracy for Information Retrieval Benchmarking. Santhanam et al. ACL 2023

---

### Author Response · Authors · 2024-11-20
**General response to the reviews**

We would like to __greatly thank all the reviewers for their valuable feedback__! To better reflect the reviewers suggestions, we made __a few updates to the text in pdf, highlighted in blue__, and __updated Figures and Tables__ with additional experiment runs suggested by the reviewers.

We would like to make some clarifications regarding the proposed approach, since we found some review summaries to be not fully reflecting or even occasionally contradicting to the essence of the proposed method.

__Our first novel contribution__ consists of casting context pruning in RAG as sequence labeling, enabling pruning of sentences _dependent on other sentences_ and the _dynamic detection_ of the number of relevant sentences (adaptability). __Due to this novel methodology, the proposed Provence outperforms prior existing baselines__ (Figure 2).

__Our second novel contribution__ is to complement reranking, an already present part of the search in RAG,  with context pruning capabilities, because these models have the same input.
Thus, instead of having a pipeline query>>retrieve>>rerank>>prune>>generate, we can do query>>retrieve>>provence>>generate, which means that __context pruning comes at almost no extra computational cost__ in the RAG pipeline.

We will also __release a model__ which __successfully performs reranking and context pruning in various domains__ (Figure 2 and Table 4) and __enables $2\times$ generation speed up in RAG for an arbitrary LLM out-of-the-box__ (Table 3).

---

### Meta-Review · Area_Chair_aQEp · 2024-12-21

**Metareview:**

This submission introduces a method for context pruning in rag o improve the inference efficiency of LLMs. The authors argue that the proposed method Provence addresses the challenges of compute overhead and irrelevant information in generated responses by dynamically removing unnecessary parts of retrieved contexts. It combines context pruning with context reranking, trains on diverse data, and proves to be efficient in real-world applications with minimal cost to RAG pipelines. The results show the proposed method works well without reducing performance and provides insights for future development of context pruners.

The strengths mentioned by the reviewers include:
- 1) the reviewers mentioned that it is well-organized;
- 2) the proposed method is tailored for the target task, with effectiveness and efficiency demonstration;
- 3) the experiments conducted are extensive.

The weaknesses include:
- 1) method-wise, during the initial review, there was concern with the question of whether developing a separate model is necessary when existing LLMs could potentially be sufficient;
- 2) Though some reviewers considered the experiments formed as its strength, the other reviewers argue that the comparisons with existing methods were insufficient and fair. The impact of hyperparameters like the retrieval pipeline and the top-k passages was not adequately addressed;
- 3) minor one: reviewers noted the lack of references to relevant prior work.

After the rebuttal, this work received mostly positive ratings (8, 8, 6, 3): 1) reviewer 99vA updated their rating to 6, 2) reviewer sJXd and 5r2e had ratings both at 8, and 3) reviewer 7vVg (rating 3) did not respond to the authors' rebuttal. Given the brevity and lack of clarity in reviewer 7vVg's review, and no engagement during the rebuttal, this review is being disregarded during the decision phase. Considering the overall ratings of 8, 8, 6 and rating increase during rebuttal, this work is recommended for an acceptance.

**Additional Comments On Reviewer Discussion:**

After the rebuttal, this work received mostly positive ratings (8, 8, 6, 3): 1) reviewer 99vA updated their rating to 6, 2) reviewer sJXd and 5r2e had ratings both at 8, and 3) reviewer 7vVg (rating 3) did not respond to the authors' rebuttal. Given the brevity and lack of clarity in reviewer 7vVg's review, and no engagement during the rebuttal, this review is being disregarded during the decision phase.

---

### Decision · Program_Chairs · 2025-01-22

Accept (Poster)